# Intrinsic Disorder as a Natural Preservative: High Levels of Intrinsic Disorder in Proteins Found in the 2600-Year-Old Human Brain

**DOI:** 10.3390/biology11121704

**Published:** 2022-11-25

**Authors:** Aaron S. Mohammed, Vladimir N. Uversky

**Affiliations:** 1Department of Molecular Medicine, Morsani College of Medicine, University of South Florida, 12901 Bruce B. Downs Blvd. MDC07, Tampa, FL 33612, USA; 2USF Health Byrd Alzheimer’s Research Institute, Morsani College of Medicine, University of South Florida, Tampa, FL 33612, USA

**Keywords:** Heslington brain, intrinsically disordered protein, intrinsically disordered region, binding-induced folding, disorder-to-order transition

## Abstract

**Simple Summary:**

Reported earlier, results of the proteomic analysis of the Heslington brain (which is at least 2600-year-old brain tissue uncovered within the skull excavated in 2008 from a pit in Heslington, Yorkshire, England) revealed the preservation of many proteins. Five of these proteins—heavy, medium, and light neurofilament proteins, glial fibrillary acidic protein, and myelin basic protein—are all engaged in the formation of non-amyloid protein aggregates, such as intermediate filaments and myelin sheath. Our bioinformatics analysis reported in this study revealed that these five proteins, their interactors, and many other proteins found in the Heslington brain are characterized by high level of intrinsic disorder, suggesting that intrinsic disorder might play a role in preserving brain tissue, likely by acting as a molecular mortar or cement that glues together various brain proteins and rigidifies the resulting assemblies, thereby generating highly stable matter.

**Abstract:**

Proteomic analysis revealed the preservation of many proteins in the Heslington brain (which is at least 2600-year-old brain tissue uncovered within the skull excavated in 2008 from a pit in Heslington, Yorkshire, England). Five of these proteins—“main proteins”: heavy, medium, and light neurofilament proteins (NFH, NFM, and NFL), glial fibrillary acidic protein (GFAP), and myelin basic (MBP) protein—are engaged in the formation of non-amyloid protein aggregates, such as intermediate filaments and myelin sheath. We used a wide spectrum of bioinformatics tools to evaluate the prevalence of functional disorder in several related sets of proteins, such as the main proteins and their 44 interactors, all other proteins identified in the Heslington brain, as well as the entire human proteome (20,317 manually curated proteins), and 10,611 brain proteins. These analyses revealed that all five main proteins, half of their interactors and almost one third of the Heslington brain proteins are expected to be mostly disordered. Furthermore, most of the remaining Heslington brain proteins are expected to contain sizable levels of disorder. This is contrary to the expected substantial (if not complete) elimination of the disordered proteins from the Heslington brain. Therefore, it seems that the intrinsic disorder of NFH, NFM, NFL, GFAP, and MBP, their interactors, and many other proteins might play a crucial role in preserving the Heslington brain by forming tightly folded brain protein aggregates, in which different parts are glued together via the disorder-to-order transitions.

## 1. Introduction

In 2008, archaeological work in advance of construction at a site on the eastern edge of Heslington village located about 1.9 miles south-east of York city center (Yorkshire, England) uncovered a skull buried in a pit. This skull belonged to an Iron Age man aged between 26–45 years old at the time of death, likely in his mid-30s. Radiocarbon dating found that the man had died sometime between 673–482 BC [1]. The most remarkable side of this discovery was the fact that several large fragments of brain were present inside the skull. Since this Heslington brain (which is at least 2600 years old) preserved many anatomical features despite being shrunk to about 20% of its original size [1], it opened a unique possibility to investigate the preservation of human brain proteins, and results of this analysis were recently reported [2]. Utilization of a broad spectrum of molecular techniques, ranging from careful exclusion of sample contamination to immunoelectron microscopy, mass-spectroscopy, quantification of brain-specific proteins by ELISA, and to gel electrophoresis and immunoblotting, demonstrated the preservation of neurocytoarchitecture in the ancient brain and also revealed the exceptional preservation of some human brain proteins in the analyzed samples of the Heslington brain [2]. Among identified proteins with this extraordinary long-term stability and ability to survive for 2600 years were proteins engaged in the formation of non-amyloid protein aggregates, such as neurofilament (NF) proteins (or NFs), glial fibrillary acidic protein (GFAP), and myelin basic protein (MBP) [2]. Based on these observations, the authors concluded that the preservation of brain for millennia was driven by the formation of protein aggregates, in which NF proteins, GFAP, and MBP played crucial roles [2]. The authors also pointed out that these proteins were strongly integrated into the aggregated mass, as, for example, it took 2–12 months to detect the resolution of intermediate filaments (IFs) from the protein aggregates [2]. Due to the likely important roles of these proteins (NF proteins, GFAP, and MBP) in the long-term preservation of the Heslington brain, they were defined as the main proteins in our study. Curiously, our analysis revealed that all these main proteins found in the Heslington brain are characterized by the high levels of intrinsic disorder.

The analysis of the micro-structure of the Heslington brain first began with immunoelectron microscopy in order to determine if there was a presence of specific axonal proteins [1,2]. Next, sensitive immunoassays were carried out to screen for other brain proteins that may have been present [2]. There were strong immune responses that indicated the presence of GFAP and MBP and weak immune responses that indicated the presence of NFs.

NF heavy (NFH), an NF subunit, was found to be present in the axons of the ancient tissue. The detection of an NF subunit indicated that other NF subunits, neurofilament middle (NFM) and neurofilament light (NFL), should be present as well. NFs are type IV intermediate filaments (IFs), which are cytoskeletal fibers found in eukaryotic neurons. Neurofilaments are necessary for the radial growth and increase of the diameter of axons [3].

GFAP is a type III IF that provides structural stability to astrocytic processes which result in the modulation of astrocyte motility and shape [4]. Being members of the IF protein family (GFAP is a type III and NFs are type IV intermediate filaments), these proteins have similar domain organization with a conserved coiled-coil central domain (which is due to the rod-like tertiary structure within the Ifs, known as rod domain) flanked by variable N-terminal “head” and C-terminal “tail” domains (or side-arms) and play crucial role in the biogenesis of the neuronal cytoskeleton. To form the neuronal cytoskeleton, these structural proteins undergo a complex, multistage assembly process that starts with the formation of α-helical coiled-coil dimers, where the central rod domains coil around each other. These dimers then assemble side-by-side to form the basic subunits of IFs, staggered tetramers [5,6,7]. On the other hand, non-helical (and highly disordered; see below) head and tail domains are needed for the IF formation, as rod domains alone do not form filaments [8]. 

MBP is a part of the insulating myelin sheath, a multilayered proteolipid membrane surrounding nerve axons, and is the second most abundant protein in the central nervous system [9]. Although MBP is a cytosolic protein, it constitutes 30% of the total myelin, being a major element of the compact myelin, in which cytoplasm is excluded, and where the cytoplasmic leaflets of myelin membrane forming the consecutive turns are practically fused, resulting in the extracellular space, which is only about 2 nm thick [10]. MBP is one of the myelin proteins that hold together a repetitive multilayer of tightly packed lipid bilayers [11]. The overall importance of myelin, this macroscopic supramolecular proteolipid structure, for the nervous system functionality is reflected in the fact that myelin proteins are among the most long-lived proteins in the body [12,13].

Mass spectroscopy was then used by Petzold et al. to get an idea of what other proteins may be present in the preserved brain [2]. Peptide fragments were detected that matched 855 proteins, after accounting for possible contaminants. GFAP, MBP, and NFL were matches for fragments that were detected; however, some of those fragments were also matches for other proteins. Only GFAP and NFL had a fragment that was unique to them. 

It was concluded by Petzold et al. that the preservation of the Heslington brain is likely due to the formation of protein aggregates [2]. The experiment that contributed most to this conclusion first involved taking samples from the white and grey matter of the preserved brain and a normal brain as the control in order to investigate the presence of protein aggregates. The samples were homogenized, centrifuged, and then the resulting liquid was subjected to gel electrophoresis. The first four bands were used for immunoblotting, which revealed the presence of hyperphosphorylated NFH with a molecular weight of 420 kDa, twice the amount to be expected. Hyperphosphorylation of NFs is thought to be one of the main triggers that lead to aggregate formation in NFs, so a large amount of hyperphosphorylated NFH is indicative that there are large protein aggregates within the preserved brain [14,15]. Another indication that the preserved brain contained large aggregates was the fact that they had a higher degree of resistance to urea than aggregates from normal brains.

Protein aggregates have a tendency to form from proteins that contain intrinsically disordered regions (IDRs), referred to as intrinsically disordered proteins (IDPs) [16]. IDPs lack the ability to spontaneously fold and form rigid 3D structures (note that some IDPs can fold under certain conditions, including upon binding to a target); they make up one third of all eukaryotic proteins [17,18]. IDRs/IDPs cannot fold because they contain a high amount of net charge and a low number of hydrophobic residues, relative to ordered proteins. A high level of net charge and low level of hydrophobicity within a given region increases repulsive forces and reduces the drive for hydrophobic collapse, respectively. IDRs allow IDPs to be very flexible and take on many different conformations. A high number of conformations allows IDPs to have numerous functions and binding partners. Although these traits are important for life, they are also the reason why IDPs are associated with a number of diseases, especially those that arise from misfolding due to certain conformational changes [16]. A higher number of possible conformations and binding partners increases the chances of misfolding taking place. Many proteins that play important roles in neurodegenerative pathways are IDPs e.g., amyloid-β, tau, and α-synuclein [19].

In this paper, intrinsic disorder and protein interactions of NFs, GFAP, and MBP are investigated. The proteins that were detected using mass spectroscopy in the Heslington brain, as well as 20,317 manually curated proteins from the entire human proteome, and 10,611 brain proteins are also analyzed to gauge the amount of intrinsic disorder that may be present within the preserved brain. Proteins that are found to be possible interactors with NFs, GFAP, and MBP are used to determine possible pathways that may have been responsible for the high degree of aggregate formation within the ancient brain.

## 2. Materials and Methods

### 2.1. Protein Datasets

ID of proteins analyzed in this study were retrieved from the rsif20190775_si_001 dataset posted by Axel Petzold, Ching-Hua Lu, Mike Groves, Johan Gobom, Henrik Zetterberg, Gerry Shaw, and Sonia O’Connor on 16.12.2019, 05:17. This dataset includes proteins listed in a Appendix A with the mass spectrometry data from [2]. Sequences of these query proteins in FASTA format were gathered from the UniProt database [20,21] and are listed in Appendix A. These retrieved sequences represent a “mass spectrometry-identified proteins” set. We conducted a comprehensive bioinformatics analysis of the neurofilament heavy, medium, and light chain proteins (NFs, UniProt IDs: P12036, P07197, P07196, respectively), glial fibrillary acidic protein (GFAP; UniProt ID: P14136), and myelin basic protein (MBP; UniProt ID: P02686), which were shown to be engaged in the formation of the non-amyloid protein aggregates [2], and which are defined here as the main proteins. The entire human proteome (20,317 manually curated proteins; proteome:UP000005640) was downloaded from the UniProt database [20,21], from which 10,611 brain proteins were further selected (proteome:UP000005640 AND brain). 

### 2.2. Evaluation of the Intrinsic Disorder Predisposition

The predispositions for intrinsic disorder of all proteins were determined using a set of commonly used per-residue disorder predictors including PONDR^®^ VLS2, PONDR^®^ VL3, PONDR^®^ VLXT, PONDR^®^ FIT, IUPred-Long, and IUPred-Short [22,23,24,25,26,27]. A web application called Rapid Intrinsic Disorder Analysis Online (RIDAO) was used to gather results from each predictor in bulk [28]. The percent of predicted intrinsic disorder residues (PPIDR) for each protein was used to classify each protein based on their levels of disorder. A residue was considered to be disordered if it had a value of 0.5 or higher. Generally, a PPIDR value of less than 10% is taken to correspond to a highly ordered protein, PPIDR between 10% and 30% is ascribed to a moderately disordered protein, and PPIDR greater than 30% corresponds to a highly disordered protein [29,30]. In addition to PPIDR, a mean disorder score (MDS) was calculated for each query protein as a protein length-normalized sum of all the per-residue disorder scores. Here again, proteins are grouped based on their corresponding MDS values, being classified as highly ordered (MDS < 0.15), moderately disordered or flexible (MDS between 0.15 and 0.5) and highly disordered (MDS ≥ 0.5).

### 2.3. PPI Networks

Protein-protein interaction (PPI) networks were generated using STRING (search tool for recurring instances of neighboring genes) platform (http://string-db.org/ accessed on 1 June 2022) [31]. In this study, STRING was used in two modes. First, a global interaction network for all 881 mass spectrometry-identified proteins was generated using a moderate confidence level of 0.7. Then, we generated networks of NFs, GFAP, and MBP using 0.7 confidence and 500 as the limit for possible number of interactors (max). 

Agile Protein Interactomes DataServer (APID, http://apid.dep.usal.es accessed on 1 June 2022) [32] was used to determine which mass spectrometry-identified proteins are involved in interactions with NFs, GFAP, and/or MBP. First, the IDs for the five main proteins were uploaded to APID. For each protein, the list of interactors with at least one item of evidence was saved, resulting in five lists of specific interactors. Each list was then compared to the list of mass spectrometry-identified proteins in order to determine if any of the interactors in the list were detected in the brain. The resulting lists were combined, and duplicates were removed. In total, 44 interactors were found. PPI networks including those interactors and the five main proteins were created using APID and STRING.

### 2.4. Disorder-Based Functional Annotations

The Database of Disordered Protein Predictions (D^2^P^2^) was used to determine binding sites based on the ANCHOR algorithm and sites of various posttranslational modifications (PTM) [33]. D^2^P^2^ also provides Structural Classification of Proteins (SCOP) domain predictions based on the SUPERFAMILY predictor and disorder predictions based on PONDR VLXT, PONDR VSL2b, PrDOS, PV2, ESpritz-DisProt, Espritz-XRay, Espritz-NMR, IUPred-Long, and IUPred-Short predictors [33]. 

### 2.5. CH-CDF Analysis

CH-CDF analysis was for gauging which proteins found through mass spectroscopy were disordered and was generated using RIDAO. This analysis combines results from charge-hydropathy (CH) and cumulative distribution function (CDF) plots, which are both binary predictors of disorder. For CH, net charge is plotted versus hydropathy for each protein [34]. Due to the observation that disordered proteins tend to have high net charge and low hydropathy, disordered and ordered proteins cluster in two regions of the plot. A linear boundary separates disordered and ordered proteins [34,35]. Proteins that are disordered appear above the boundary while ordered proteins appear below [34,35]. In the CDF-plot predictor, PONDR scores for each residue of a single protein are plotted against their frequency within the sequence. If a CDF curve of a given protein is below the order-disorder boundary, this protein is considered to be disordered, and ordered if the CDF curve is located above this boundary [35]. Data generated by CH- and CDF-plots are then combined to generate a ΔCH-ΔCDF plot [36,37,38], which enables rapid discrimination between flavors of disorder [39]. To this end, for each query protein, ΔCH, the vertical distance of the corresponding point in CH plot from the boundary, is calculated, whereas ΔCDF is computed as the average distance between the order-disorder boundary and the CDF curve. Then, ΔCH is plotted against ΔCDF, resulting in a CH-CDF plot. Proteins in the top-left quadrant are predicted to be disordered by both CH and CDF, the ones in the bottom-left are predicted to be ordered by CH and disordered by CDF, the ones in the top-right are predicted to be disordered by CH and ordered by CDF, and the ones in the bottom-right quadrant are predicted to be ordered by both [36,37,38,39]. 

### 2.6. Statistical Analysis

The statistical significance of the differences in the disorder status of different protein groups analyzed in this study was evaluated using the *p*-value calculator accessible at the (https://www.gigacalculator.com/calculators/p-value-significance-calculator.php accessed on 20 October 2022). Z-test statistics were used in these analyses. 

### 2.7. Pathway Analysis

The gene IDs of the 44 proteins that were found to interact with NFs, GFAP, and/or MBP were inputted into a webserver called DAVID, which stands for Database for Annotation, Visualization, and Integrated Discovery (http://www.david.niaid.nih.gov accessed on 1 June 2022) [40]. It was used to investigate if any of the 44 interactors that reside in the preserved brain are part of any pathways that involve protein aggregate formation. The five proteins were also uploaded to the KEGG (Kyoto Encyclopedia of Genes and Genomes) [41] pathway database (https://www.genome.jp/kegg/pathway.html accessed on 1 June 2022) to carry out their functional analysis. 

### 2.8. 3D Model Structures of Main Proteins

3D structural models of the main proteins (NFs, GFAP, and MBP) were generated by AlphaFold [42].

## 3. Results

### 3.1. Functional Intrinsic Disorder in the Main Proteins

At the first stage of our study, we looked at the intrinsic disorder predisposition of five main proteins, the neurofilament heavy, medium, and light chain proteins (NFH, NFM, and NFL, UniProt IDs: P12036, P07197, P07196, respectively), glial fibrillary acidic protein (GFAP; UniProt ID: P14136), and myelin basic protein (MBP; UniProt ID: P02686), which were shown to be engaged in the formation of the non-amyloid protein aggregates, and which were considered as the major constituents defining the preservation of the Heslington brain for at least 2600 years [2]. Results of the multifactorial disorder analysis of these natural preservatives are shown in Figure 1, Figure 2, Figure 3, Figure 4 and Figure 5. 

For these five main proteins, we looked at the per-residue disorder profiles generated by RIDAO, functional disorder profiles produced by the D2P2 platform, protein-protein interaction networks generated by STRING, and 3D structural models generated by AlphaFold. 

#### 3.1.1. NFH 

Figure 1 represents results of a functional disorder analysis of human NFH, which is a 1026 residue-long intermediate filament (IF) protein that together with the NFM and NFL is involved in the maintenance of the neuronal caliber. This protein includes a head region (residues 1–200), an UF rod domain (residues 97–413) and a tail region (residues 414–1026) containing 30 × 6 amino acid repeats of K-S-P-[AEPV]-[EAK]-[AEVK]. Many serines of the K-S-P repeats of the NFH protein are phosphorylated in addition to Ser76, Ser124, Ser347, and Thr774. It is believed that the NFH phosphorylation within the repeat region leads to the formation of interfilament cross bridges that play important roles in the maintenance of axonal caliber [43], whereas phosphorylation within the head and rod regions inhibits polymerization [43]. 

**Figure 1 biology-11-01704-f001:**
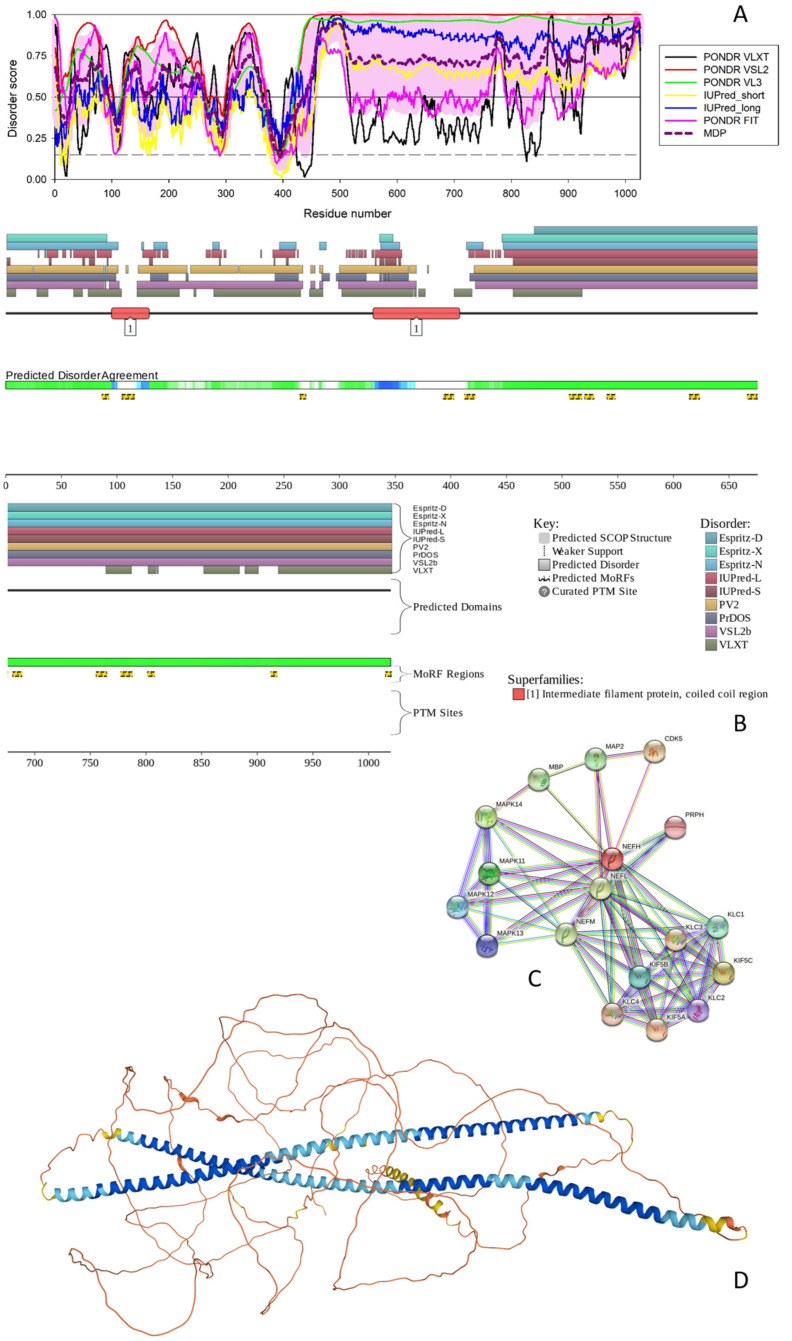
Functional disorder analysis of human NFH (UniProt IDs: P12036). (**A**) Per-residue disorder profile generated by RIDAO. Solid and dashed horizontal lines at disorder scores 0.5 and 0.15 correspond to the disorder and flexibility thresholds. (**B**) Functional disorder profile produced by the D^2^P^2^ platform. Here, the IDR localization predicted by IUPred, PONDR^®^ VLXT, PONDR^®^ VSL2, PrDOS, PV2, and ESpritz are shown by nine differently colored bars on the top of the plot, whereas the blue-green-white bar in the middle of the plots shows the agreement between the outputs of these disorder predictors, with disordered regions by consensus being shown by blue and green. The two lines with colored and numbered bars above the disorder consensus bar show the positions of functional SCOP domains [44,45] predicted using the SUPERFAMILY predictor [46]. Positions of the predicted disorder-based binding sites (MoRF regions) identified by the ANCHOR algorithm are shown by yellow zigzagged bars [47]. Locations of the sites of different posttranslational modifications (PTMs) identified by the PhosphoSitePlus platform [48] are shown at the bottom of the plot by the differently colored circles. (**C**) Protein-protein interaction (PPI) network of human NFH generated by STRING using seven types of evidence shown by differently colored lines: a black line represents co-expression evidence; a blue line—co-occurrence evidence; a green line—neighborhood evidence; a light blue line—database evidence; a purple line—experimental evidence; a red line—the presence of fusion evidence; and a yellow line—text-mining evidence [31]. (**D**) 3D structure modeled for NFH by AlphaFold. Structure is colored based on the pLDDT values, where orange, yellow, cyan, and blue colors correspond to the segments predicted by AlphaFold with very high very low (pLDDT < 50), low (70 > pLDDT > 50), high (90 > pLDDT > 70), and (pLDDT > 90) confidence.

**Figure 2 biology-11-01704-f002:**
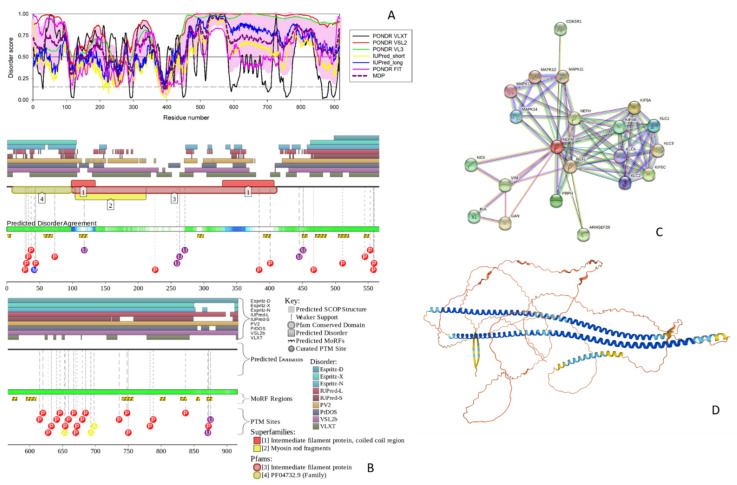
Functional disorder analysis of human NFM (UniProt IDs: P07197). (**A**) Per-residue disorder profile generated by RIDAO. (**B**) Functional disorder profile produced by the D2P2 platform. (**C**) STRING-generated PPI network centered at NFM. (**D**) 3D structure modeled for NFM by AlphaFold. Structure is colored based on the pLDDT values, where segments predicted with very high (pLDDT > 90), high (90 > pLDDT > 70), low (70 > pLDDT > 50), and very low (pLDDT < 50) confidence are shown by blue, cyan, yellow and orange colors, respectively.

**Figure 3 biology-11-01704-f003:**
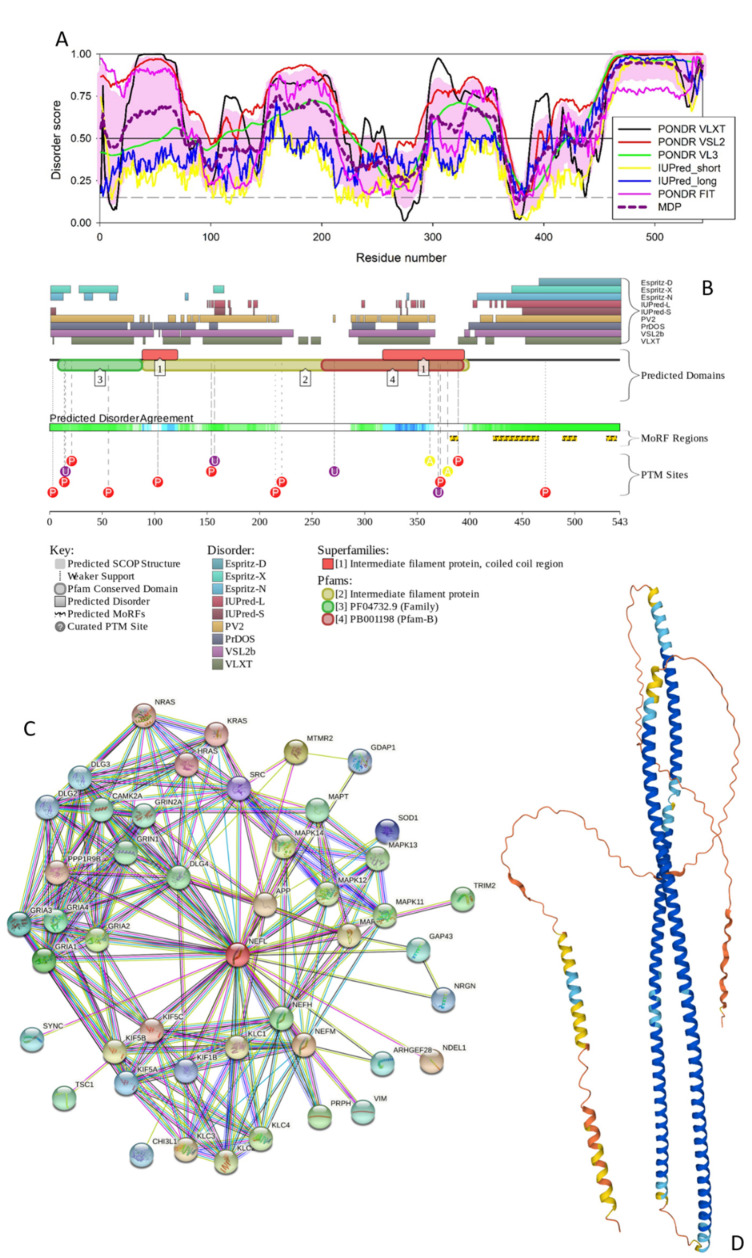
Functional disorder analysis of human NFL (UniProt IDs: P12036). (**A**) Per-residue disorder profile generated by RIDAO. (**B**) Functional disorder profile produced by the D2P2 platform. (**C**) STRING-generated PPI network centered at NFL. (**D**) 3D structure modeled for NFL by AlphaFold. Structure is colored based on the pLDDT values, where segments predicted with very high (pLDDT > 90), high (90 > pLDDT > 70), low (70 > pLDDT > 50), and very low (pLDDT < 50) confidence are shown by blue, cyan, yellow and orange colors, respectively.

**Figure 4 biology-11-01704-f004:**
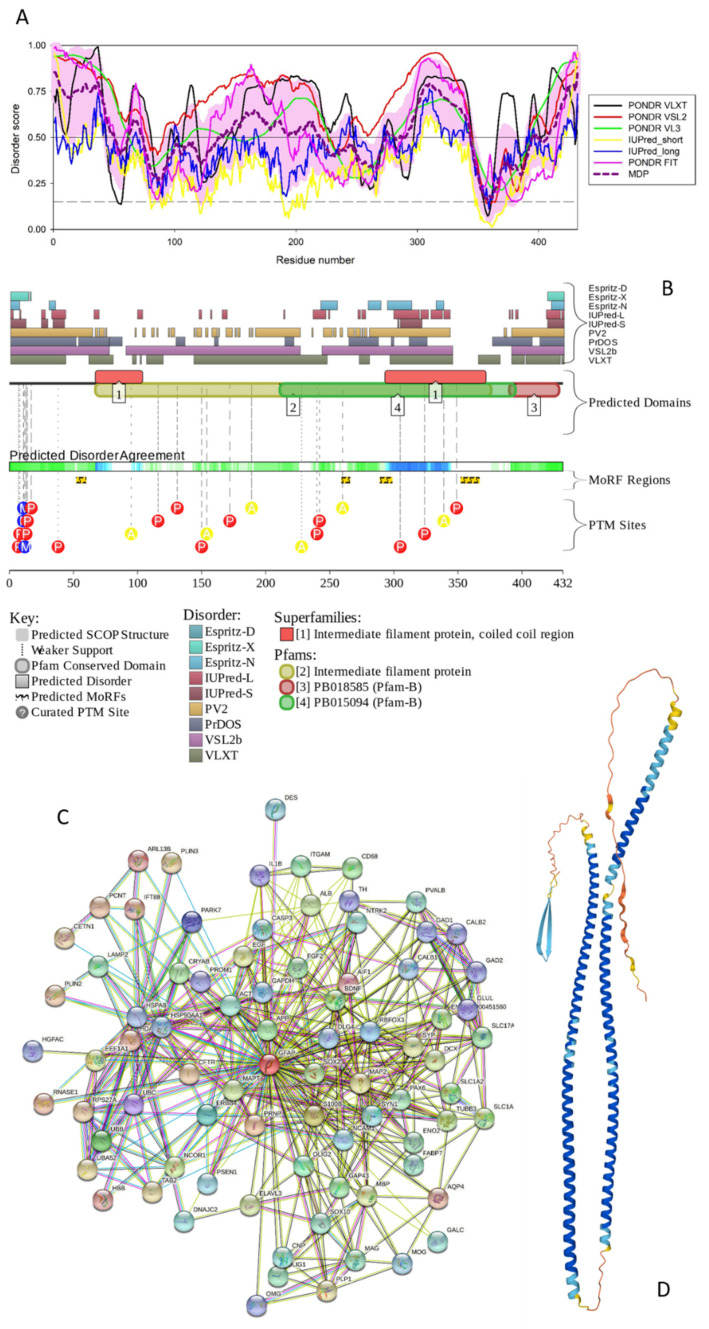
Functional disorder analysis of human glial fibrillary acidic protein (GFAP; UniProt ID: P14136). (**A**) Disorder profile generated by RIDAO. (**B**) D^2^P^2^ platform-generated functional disorder. (**C**) STRING-generated PPI network centered at GFAP. (**D**) 3D structure modeled for GFAP by AlphaFold. Structure is colored based on the pLDDT values, where segments predicted with very high (pLDDT > 90), high (90 > pLDDT > 70), low (70 > pLDDT > 50), and very low (pLDDT < 50) confidence are shown by blue, cyan, yellow and orange colors, respectively.

**Figure 5 biology-11-01704-f005:**
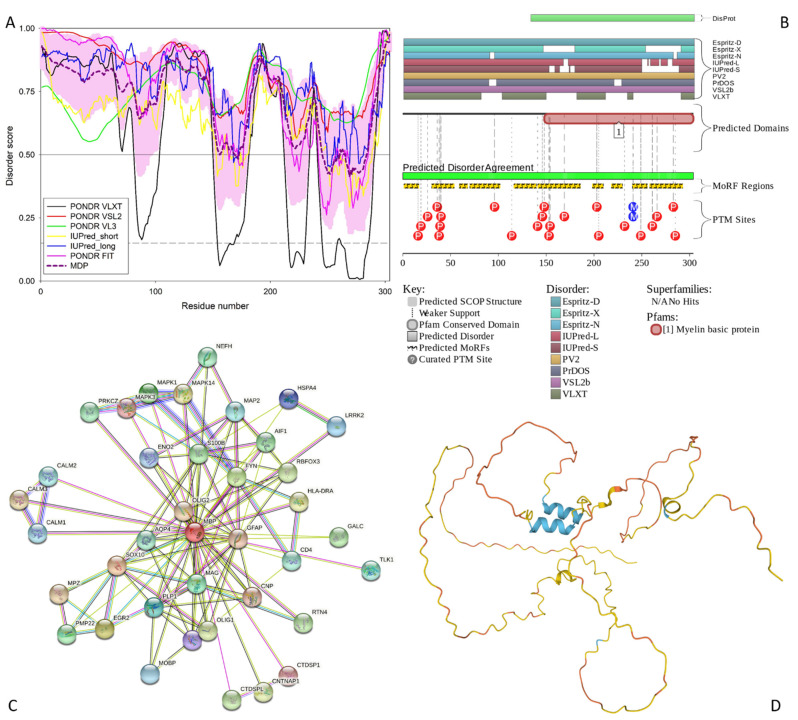
Functional disorder analysis of the myelin basic protein (MBP; UniProt ID: P02686). (**A**) Per-residue disorder profile generated by RIDAO. (**B**) Functional disorder profile produced by the D^2^P^2^ platform. (**C**) STRING-generated PPI network centered at MBP. (**D**) 3D structure modeled for MBP by AlphaFold. Structure is colored based on the pLDDT values, where segments predicted with very high (pLDDT > 90), high (90 > pLDDT > 70), low (70 > pLDDT > 50), and very low (pLDDT < 50) confidence are shown by blue, cyan, yellow and orange colors, respectively.

The PPIDR values generated by the PONDR^®^ FIT, PONDR^®^ VSL2, PONDR^®^ VL3, PONDR^®^ VLXT, IUPred Short, and IUPred Long were 56.34%, 90.45%, 85.77%, 50.19%, 58.87%, and 71.05%, respectively. From the mean of these predictors, 83.82% of the NFH residues were predicted to be disordered (scores greater than or equal to 0.5). Based on these parameters, NFH can be classified as a highly disordered protein. The most disorder is found between residue 440 and the C-terminal tail (see Figure 1A). NFH exists as two isoforms generated by alternative splicing, with isoform 2 being different from the canonical form by missing residues 750–812 within the tail region. 

The ANCHOR algorithm identified 16 disorder-based binding sites (i.e., IDRs capable of the disorder-to-order transition at interactions with specific partners and known as molecular recognition features, MoRFs) in NFH: 86–92, 104–115, 264–269, 393–402, 412–421, 506–517, 520–528, 540–547, 614–623, 666–675, 680–688, 755–764, 777–787, 801–807, 912–917, and 1015–1020 (see Figure 1B). These indicate that NFH can be engaged in multiple disorder-driven interactions. This conclusion is supported by Figure 1C, showing the PPI network generated by STRING using 0.7 as the confidence level. There are 18 interactors and 73 interactions in this PPI network (see Figure 1C). This network centered at NFH has significantly more interactions than expected, since in a random set of proteins of the same size and degree distribution drawn from the human proteome one would expect to find 18 interactions (the PPI enrichment *p*-value of <10^−16^). Furthermore, the average local clustering coefficient of this network is 0.856, and its average node degree is 8.11, indicating that on average, each protein in the network interacts with at least eight partners.

All these data indicate that the NFH-centered PPI network is highly connected. The most enriched biological processes of the proteins in this network (in terms of the Gene Ontology, GO) are Microtubule-based process (GO:0007017; *p* = 2.29 × 10^−6^), Axo-dendritic transport (GO:0008088; *p* = 2.88 × 10^−6^), Neurofilament bundle assembly (GO:0033693; *p* = 5.28 × 10^−5^), Microtubule-based movement (GO:0007018; *p* = 5.28 × 10^−5^), and Anterograde dendritic transport of neurotransmitter receptor complex (GO:0098971; *p* = 5.85 × 10^−5^). Among the five most enriched molecular functions are Microtubule motor activity (GO:0003777; *p* = 1.72 × 10^−9^), Microtubule binding (GO:0008017; *p* = 2.26 × 10^−6^), MAP kinase activity (GO:0004707; *p* = 4.25 × 10^−6^), Cytoskeletal protein binding (GO:0008092; *p* = 3.36 × 10^−5^), and Kinesin binding (GO:0019894; *p* = 4.02 × 10^−5^). Finally, the most enriched cellular components related to the proteins in this network are Polymeric cytoskeletal fiber (GO:0099513; *p* = 2.39 × 10^−12^), Kinesin complex (GO:0005871; *p* = 4.54 × 10^−11^), Microtubule associated complex (GO:0005875; *p* = 2.31 × 10^−10^), Axon (GO:0030424; *p* = 4.06 × 10^−10^), and Microtubule (GO:0005874; *p* = 7.04 × 10^−9^. 

Finally, Figure 1D shows that most of the NFH 3D structure is predicted by AlphaFold with low or very low per-residue confidence scores (70 > pLDDT > 50 and pLDDT < 50), indicating that most of this protein may be unstructured in isolation. Furthermore, it is very likely that the two long α-helices (residues 90–248 and 267–410), which are predicted by AlphaFold and overlap with the coil 1A/Bb (residues 101–132/146–244) and coil 2A/2B (residues 267–288/293–413) regions, will be formed as a result of the coiled-coil-based assembly of the neuronal intermediate filaments, where “three polypeptides first make α-helical coiled-coil dimers, then form antisymmetric tetramers and finally assemble into 10–15 nm diameter filaments with long flexible side-arms extending 50–100 nm from the polymer core” [49]. It was also indicated that initial steps of the neurofilament assembly rely on the preferential NFH-NFL and NFM-NFL heterodimerization [50,51]. Since in the mature axons, the stoichiometry of NFL:NFM:NFH is 7:3:2, on average, one can find there 1 NFL:NFL homodimer, 2 NFH:NFL dimers, and 3 NFM:NFL dimers [49]. 

It is important to note here that long α-helical segments found in NFH, as well as NFH, NFL, and GFAP (see section below), correspond to their rod domains, which are engaged in the formation of highly intertwined coiled-coil dimers. Clearly, these long α-helical segments cannot exist in isolation, as there are no physical force(s) that would be able to stabilize such a structure in aqueous media. However, these long α-helical segments are easily formed and stabilized at the coiled-coil formation. They clearly represent an extreme case of the folding-induced binding phenomenon described for many IDPs. Furthermore, coiled-coils are often cross-predicted to be unstructured regions [52,53]. Therefore, the structural models generated for long coiled-coil segments of these (and many other) proteins by AlphaFold reflect the potential of this AI platform (which is a machine learning approach incorporating multi-sequence alignments, as well as physical and biological information on protein structure, into a deep learning algorithm [42])) to find such disordered regions with high folding-upon-binding potential. In agreement with these considerations, a recent report indicated that AlphaFold2 can capture the folded state structure of IDRs known to fold under specific conditions [54]. In fact, the authors of this preprint indicated: “AlphaFold2 assigns high-confidence scores to about 60% of a set of 350 IDRs that have been reported to conditionally fold, suggesting that AlphaFold2 has learned to identify conditionally folded IDRs, which is unexpected, since IDRs were minimally represented in the training data” [54]. Therefore, in addition to be able to accurately identify ordered domains and IDRs [55,56,57,58], AlphaFold is capable of finding IDRs that conditionally fold [54].

#### 3.1.2. NFM 

NFM (UniProt ID: P07197) is a 916-residue long protein. Similar to NFH, this protein has a head region (residues 2–104), IF rod domain (residues 101–412) containing coiled-coil regions 1A, 1B, 2A, and 2B (residues 105–136, 150–248, 266–287, and 292–412, respectively), and a tail region containing 6 × 13 AA approximate tandem repeats of K-S-P-V-[PS]-K-S-P-V-E-E-[KA]-[GAK]. The isoform 2 of NFM is different from the canonical form by missing a head and most of the rod regions (residues 1–376). 

Also similar to NFM, this intermediate filament protein is phosphorylated on multiple serine regions within the repeats of the tripeptide K-S-P and in the head and rod domains. In addition to containing multiple phosphoserines, NFM is phosphorylated on Thr571, contains two O-linked (GlcNAc) threonines at positions 47 and 431, and includes Omega-N-methylarginine at position 42 (https://www.uniprot.org/uniprotkb/P07197/ accessed on 1 June 2022). 

The PPIDR values according to each disorder predictor suggest that the NFM is a highly disordered protein. For PONDR^®^ FIT, PONDR^®^ VSL2, PONDR^®^ VL3, PONDR^®^ VLXT, IUPred Short, and IUPred Long, the PPIDR values were 58.52%, 88.32%, 84.83%, 56.44%, 45.20%, and 64.74%, respectively. The PPIDR from the mean of these predictors was 77.62%. According to the mean, the disordered regions include residues 1–106, 160–213, 273–282, 303–364, 430–441, and 454 to the C-terminus (see Figure 2A). According to the ANCHOR algorithm, NFM has 16 disorder-based binding sites: 1–6, 56–69, 110–122, 290–299, 390–401, 449–457, 469–486, 504–520, 543–551, 573–580, 594–609, 740–757, 798–806, 829–838, 853–858, and 869–877 (Figure 2B). 

The STRING-generated PPI network, shown in Figure 2C, includes 21 interactors, 77 interactions, and is characterized by an average node degree of 7.33 and an average local clustering coefficient of 0.88. Therefore, this NFM-centered network has significantly more interactions than expected (the PPI enrichment *p*-value of <10^−16^), since in a random set of proteins of the same size and degree distribution drawn from the human proteome one would expect to find 21 interactions. 

Among the most enriched biological processes related to proteins in this network are Intermediate filament cytoskeleton organization (GO:0045104; *p* = 2.50 × 10^−7^), Axo-dendritic transport (GO:0008088; *p* = 4.52 × 10^−5^), Neurofilament bundle assembly (GO:0033693; *p* = 6.81 × 10^−5^), Anterograde dendritic transport of neurotransmitter receptor complex (GO:0098971; *p* = 6.81 × 10^−5^), and Intermediate filament organization (GO:0045109; *p* = 6.81 × 10^−5^). The most enriched molecular functions of these proteins are Microtubule motor activity (GO:0003777; *p* = 6.23 × 10^−9^), MAP kinase activity (GO:0004707; *p* = 1.10 × 10^−5^), Structural constituent of postsynaptic intermediate filament cytoskeleton (GO:0099184; *p* = 1.72 × 10^−5^), Tubulin binding (GO:0015631; *p* = 4.68 × 10^−5^), and Structural constituent of cytoskeleton (GO:0005200; *p* = 4.98 × 10^−5^). These proteins were most abundantly found within the following cellular components: Polymeric cytoskeletal fiber (GO:0099513; *p* = 1.11 × 10^−12^), Supramolecular fiber (GO:0099512; *p* = 1.11 × 10^−12^), Kinesin complex (GO:0005871; *p* = 6.73 × 10^−11^), Cytoskeleton (GO:0005856; *p* = 2.62 × 10^−9^), and Neurofilament (GO:0005883; *p* = 1.02 × 10^−7^).

The high level of disorder in this protein is further evidenced by Figure 2D, which shows its 3D structure modeled by AlphaFold, where one can see two very long and intertwined α-helices (residues 92–250 and 267–416), a shorter α-helix (residues 453–492), and a β-hairpin (residues 851–883), with the remaining protein being expected to be unstructured in isolation. Again, with a high probability, long helical segments are formed as a result of the coiled-coil-driven dimerization of NFM with NFL.

#### 3.1.3. NFL

Human NFL (UniProt ID: P07196) is the shortest intermediate filament protein and contains 545 residues. Despite being noticeably shorter, this protein contains a head and IF rod domains (residues 2–92 and 90–400, respectively), and a tail region (residues 397–543) with the subdomains A and B (residues 397–443 and 444–543, respectively). Moreover, there are coils 1A (residues 93–124), 1B (residues 138–234), 2A (residues 253–271), and 2B (residues 281–396) in the rod domain. NFL is phosphorylated in the head and rod regions, and contains O-linked (GlcNAc) threonine and O-linked (GlcNAc) serine at positions 21 and 30, respectively, and Omega-N-methylarginine at position 30. The Arg23 residue of this protein can be modified to Asymmetric dimethylarginine or Omega-N-methylarginine (https://www.uniprot.org/uniprotkb/P07196/entry#ptm_processing accessed on 1 June 2022). 

For this protein, the PPIDR values evaluated from the PONDR^®^ VSL2, PONDR^®^ VLXT, PONDR^®^ VL3, PONDR^®^ FIT, IUPred, and Long IUPred Short outputs were 82.69%, 67.40%, 60.22%, 58.38%, 28.73%, and 20.81%, respectively, whereas the PPIDR from the mean disorder predictor was 56.72%, indicating that, similar to other intermediate filament proteins, NFL is expected to be highly disordered (see Figure 3A). According to the mean disorder profile (see Figure 3A), the disordered regions of NFL include residues 1–7, 20–75, 147–213, 295–359, 416–424, and 441–545. 

Figure 3B shows that NFL is predicted by ANCHOR to have four disorder-based binding sites (MoRF regions): 381–388, 422–465, 488–501, and 530–539 and contains multiple PTMs. 

The NFL-centered PPI network shown in Figure 3C includes 46 interactors, 214 interactions, which significantly exceeds the expected number of edges of 63. This highly enriched network (PPI enrichment *p*-value of <10^−16^) is characterized by the average node degree of 9.3 and an average local clustering coefficient of 0.77. 

Proteins in this network are enriched in the following biological processes: Regulation of NMDA receptor activity (GO:2000310; *p* = 6.82 × 10^−16^), Regulation of neurotransmitter receptor activity (GO:0099601; *p* = 8.97 × 10^−15^), Axo-dendritic transport (GO:0008088; *p* = 6.23 × 10^−14^), Modulation of chemical synaptic transmission (GO:0050804; *p* = 3.94 × 10^−3^), and Regulation of cation channel activity (GO:2001257; *p* = 2.27 × 10^−11^). The most enriched molecular functions are Ionotropic glutamate receptor activity (GO:0004970; *p* = 5.40 × 10^−8^), Microtubule motor activity (GO:0003777; *p* = 5.40 × 10^−8^), Microtubule binding (GO:0008017; *p* = 3.96 × 10^−7^), AMPA glutamate receptor activity (GO:0004971; *p* = 9.93 × 10^−7^), and Transmitter-gated ion channel activity involved in regulation of postsynaptic membrane potential (GO:1904315; *p* = 2.25 × 10^−6^). Finally, among the most enriched cellular components, there are Neuron projection (GO:0043005; *p* = 2.17 × 10^−23^), Axon (GO:0030424; *p* = 1.41 × 10^−20^), Somatodendritic compartment (GO:0036477; *p* = 5.13 × 10^−18^), Postsynaptic density (GO:0014069; *p* = 2.19 × 10^−17^), and Dendrite (GO:0030425; *p* = 2.19 × 10^−17^). 

As per AlphaFold, human NFL is expected to contain two long intertwined α-helices (residues 73–241 and 255–403) characteristic of the coiled-coil structures with a shorter C-terminal helix (residues 481–539) predicted with low confidence (see Figure 3D). This is a typical structural organization of the proteins involved in the formation of intracellular skeletons and extracellular matrices. It is likely that such long intertwined helices are formed as a result of the binding-induced folding, as extremely long helices cannot exist in isolation. In line with these considerations, it was noted that compared to the complete human proteome, the extracellular proteome is significantly enriched in proteins with high disorder content (>50%) and includes many coiled-coil proteins [52]. 

Furthermore, it was indicated that coiled-coils are commonly predicted to be unstructured regions [53]. Therefore, it is not surprising that human NFL is expected to show high helical content despite being predicted to be mostly disordered protein. Curiously, NFL is the only IF protein that can form homodimers [59] and is also crucial for the formation of heterodimers with NFM and NFH [50,51]. Therefore, the intrinsically disordered nature of these three IF proteins and their capability to undergo binding-induced disorder-to-order transition leading to the formation of long α-helical segments are crucial for the neurofilament assembly. 

#### 3.1.4. GFAP

Human glial fibrillary acidic protein (GFAP; UniProt ID: P14136) is a type III intermediate filament protein that serves as a highly specific marker for the central nervous system (CNS) astrocytes [60,61,62] and which is most abundantly expressed in the brain [63]. In addition to GFAP, a group of type III intermediate filament protein includes desmin, peripherin, and vimentin [64]. A very specific feature of GFAP is the presence of multiple isoforms generated by alternative splicing and alternative polyadenylation, where in humans, in addition to the canonical isoform GFAPα, there are twelve splice-isoforms [63,65,66,67]. The domain structure of GFAP is typical for the intermediate filament proteins, and this protein contains a head domain (residues 1072), an IF rode (residues 69–377) that includes coils 1A (residues 73–104), 1B (residues 116–214), 2A (residues 231–252), and 2B (residues 257–377), and the tail region (residues 378–432). Human GFAP contains multiple PTMs, being phosphorylated at Thr7, Ser13, Ser38, Ser82, Thr110, Thr150, Ser323, Thr383, and Ser385 [68,69]. It also citrullinates on arginine residues at positions 30, 36, 270, 406, and 416 [70], and has an Omega-N-methylarginine at position 12. Figure 4A shows that human GFAP is predicted to be a highly disordered protein. 

In fact, the PPIDR values derived for GFAP from PONDR^®^ VSL2, PONDR^®^ VLXT, PONDR^®^ VL3, PONDR^®^ FIT, IUPred Long, and IUPred Short were 82.41%, 68.52%, 59.72%, 48.84, 22%, and 12.96%, respectively, whereas the PPIDR based on the mean disorder prediction was 55.09%, thereby classifying GFAP as a highly disordered protein. Looking at the mean disorder profile (see Figure 4A), one can find seven disordered regions in this protein: residues 1–51, 63–73, 136–217, 274–281, 291–343, 395–398, and 408–432. There are four disorder-based binding sites in GFAP at residues 52–59, 259–265, 289–298, and 352–366 (see Figure 4B). The D^2^P^2^-based functional disorder profile also shows that the human GFAP is densely decorated by various PTMs (see Figure 4B). 

Figure 4C represents the STRING-generated PPI network with the average local clustering coefficient of 0.646 and the average node degree of 10.4. This GFAP-centered network contains 81 proteins engaged in 420 interactions, which is significantly more than the expected 159 interactions (PPI enrichment *p*-value of <10^−16^). Analysis of the functional enrichment of the proteins included in this network (in terms of the GO terms) revealed that the five most enriched biological processes are Nervous system development (GO:0007399; *p* = 1.50 × 10^−17^), Neuron differentiation (GO:0030182; *p* = 2.71 × 10^−15^), Central nervous system development (GO:0007417; *p* = 8.51 × 10^−15^), Neuron development (GO:0048666; *p* = 9.01 × 10^−14^), and Generation of neurons (GO:0048699; *p* = 1.97 × 10^−13^). The most enriched molecular functions of these proteins are Protein binding (GO:0005515; *p* = 3.76 × 10^−10^), Binding (GO:0005488; *p* = 3.96 × 10^−6^), Identical protein binding (GO:0042802; *p* = 0.00012), Protein tag (GO:0031386; *p* = 0.00014), and Enzyme binding (GO:0019899; *p* = 0.00014). Among the most common cellular components are 3.07.40 × 10^−20^), Somatodendritic compartment (GO:0036477; *p* = 1.64 × 10^−17^), Neuron projection (GO:0043005; *p* = 1.64 × 10^−17^), Plasma membrane bounded cell projection (GO:0120025; *p* = 1.64 × 10^−17^), and Cell projection (GO:0042995; *p* = 1.64 × 10^−17^).

Figure 4D represents the 3D structure modeled for GFAP by AlphaFold and shows that this protein is structurally similar to other IF proteins and contains two long intertwined α-helices (residues 52–106/110–215 and 233–378). With a high probability, these structural elements are formed as a result of the binding-induced folding. In fact, similar to the neurofilament assembly discussed in sections dedicated to NFH, NFM and NFL, GFAP forms an IF network, where “GFAP can form homodimers, but also heterodimers with vimentin. In the next step of IF network assembly, dimers bind in an antiparallel fashion to form tetramers, which then laterally associate into octamers forming structures called unit-length fragments (ULFs). Subsequent association of ULFs in a non-polar fashion leads to the formation of a filament, which then undergoes radial compaction leading to the final diameter of 10 nm” [63,71]. 

#### 3.1.5. MBP

Human myelin basic protein (MBP; UniProt ID: P02686) is a 304-residue-long intrinsically disordered protein [72], which is highly abundant in the central nervous system (CNS) myelin sheath, being crucial for the maintenance of the structural integrity of this macroscopic supramolecular proteolipid structure that wraps around the axon in a spiral fashion [11]. MBP ensures tight, spiral, multilamellar arrangement of the myelin sheath by holding together the apposing cytoplasmic leaflets of the oligodendrocyte membrane in a [73]. This protein is encoded by an *MBP* gene, which is included into the Golli (Genes of the Oligodendrocyte Lineage) gene complex and which contains 11 exons in mice and 10 in humans, including the seven exons translated into the classic MBP [74]. However, MBP is known to exist as several isoforms, which are noticeably different in size and charge (e.g., 21.5, 20.2, 18.5, 17.24, 17.22, and 14 kDa in the mouse and 21.5, 20.2, 18.5, and 17.2 kDa in humans) and which are generated by alternative splicing of a single mRNA [9,72,75,76,77,78]. Being characterized by a very basic pI (pI 9.79), the longest MBP isoform (Golli-MBP1 or HOG7; 304 residues, 33.1 kDa) contains 29 Arg, 19 Lys, 22 Asp, and 16 Glu residues. On the other hand, the alternative splicing-generated isoforms 2 (Golli-MBP2 or HOG5; 197 residues, 21.5 kDa), 3 (MBP1; 197 residues, 21.5 kDa), 4 (MBP2; 186 residues, 20.2 kDa), 5 (MBP3, 171 residues, 18.5 kDa), and 6 (MBP4, 160 residues, 17.2 kDa) are characterized by the pI values of 5.99, 11.35, 11.14, 11.38, and 11.14, respectively. Furthermore, MBP isolated from brains shows extensive PTMs, such as deimination, phosphorylation, deamidation, methylation, and N-terminal acylation [79,80]. Although no specific functional domains were assigned to MBP, this protein contains several regions with characteristic amino acid biases, such as regions enriched in basic and acidic residues (1–44 and 95–117) or polar residues (45–71 and 118–147) (https://www.uniprot.org/uniprotkb/P02686/entry accessed on 1 June 2022). 

The intrinsically disordered nature of the MBP has been known for more than 50 years [81], being confirmed over and over again in multiple comprehensive studies [73,82,83,84], including an impressive (but unsuccessful) attempt to crystallize MBP under 4600 different conditions that eventually resulted in an important conclusion that “18.5 kDa MBP and possibly also its isoforms will remain preeminent examples of proteins that cannot be crystallized” [85]. In agreement with these studies, our computational analysis revealed that the human MBP is an extremely disordered protein. In fact, the PPIDRs calculated using the outputs of PONDR^®^ FIT, PONDR^®^ VSL2, PONDR^®^ VL3, PONDR^®^ VLXT, IUPred Short, and IUPred Long were 82.57%, 100%, 100%, 58.22%, 82.57%, and 93.75%, respectively (see Figure 5). 

The PPIDR evaluated from the mean disorder profile was 86.51%. Therefore, out of the five main proteins analyzed in this study, MBP displayed the most disorder (see Figure 5A). The disordered regions include residues 1–154, 157–168, 173–245, 261–264, and 285–304. Importantly, this highly disordered nature seems to be crucial for the MBP functionality, as this protein is heavily decorated by diverse PTMs, such as multiple phosphorylation, citrullination, and methylation sites (see Figure 5B), many of which are isoform-specific [9]. Furthermore, MBP is predicted by the ANCHOR to contain 10 MoRFs: residues 1–16, 30–53, 59–67, 70–101, 116–139, 141–185, 198–209, 218–229, 240–255, and 258–292 (see Figure 5B). Therefore, 67.4% of the MBP residues are potentially involved in disorder-based interactions with binding partners. 

This is reflected in the well-developed PPI network generated for human MBP by STRING (see Figure 5C), which contains 38 interactors and 112 interactions and is characterized by the average local clustering coefficient of 0.791 and average node degree of 5.89. Since a set of proteins of the same size and degree distribution randomly drawn from the human proteome is expected to have 46 interactions, the MBP-centered network has significantly more interactions than expected (PPI enrichment value of 3.33 × 10^−16^). Proteins involved in the formation of this network commonly participate in the following biological processes: Nervous system development (GO:0007399; *p* = 2.08 × 10^−15^), Central nervous system development (GO:0007417; *p* = 5.74 × 10^−14^), Myelination (GO:0042552; *p* = 7.53 × 10^−11^), Gliogenesis (GO:0042063; *p* = 7.53 × 10^−11^), and System development (GO:0048731; *p* = 7.53 × 10^−11^). The most enriched molecular functions of these proteins are N-terminal myristoylation domain binding (GO:0031997; *p* = 0.00028), MAP kinase kinase activity (GO:0004708; *p* = 0.00028), Adenylate cyclase activator activity (GO:0010856; *p* = 0.00040), Disordered domain specific binding (GO:0097718; *p* = 0.00068), and Structural constituent of myelin sheath (GO:0019911; *p* = 0.0011). Finally, these proteins are most commonly found in the following cellular components: Myelin sheath (GO:0043209; *p* = 1.50 × 10^−10^), Cell body (GO:0044297; *p* = 9.55 × 10^−6^), Synapse (GO:0045202; *p* = 9.55 × 10^−6^), Neuronal cell body (GO:0043025; *p* = 1.88 × 10^−5^), and Somatodendritic compartment (GO:0036477; *p* = 0.00015).

In line with the outputs of our bioinformatics analysis and in line with the available experimental data, Figure 5D shows that human MBP is predicted by the AlphaFold as a highly disordered protein. Curiously, MBP regions predicted by AlphaFold to contain an α-helical structure (residues 170–179, 219–228, and 283–287) coincide with or are included in the corresponding MoRFs (residues 141–185, 218–229, and 258–292), indicating that these structural elements predicted by AlphaFold might correspond to the functional configurations of these MBP segments that are likely to fold at interaction with specific partners. Furthermore, NBP shows a strong propensity for self-association [86,87,88,89,90,91,92], and this self-association is enhanced by detergents and lipids [89]. In addition to interaction with proteins, MBP, being classified as a peripheral membrane protein, can bind to lipid membranes [93] and undergo pronounced conformational changes [86,94], e.g., based on the low-resolution structural analyses, a pronounced transition from the mostly disordered form to a conformation with the noticeably increased content of α-helical and β-sheet structure induced in MBP by phosphorylation and in the presence of detergents, lipids, and organic solvents [86,95,96,97,98,99]. 

#### 3.1.6. The Main Proteins as Illustrations of the “Stability of Instability” Phenomenon

It is known that in addition to being responsible for the formation of cellular scaffolds of neurons, under some pathological conditions IF proteins can self-assemble into intra- and extracellular aggregates [3,100,101,102,103,104,105,106]. For example, Lewy-body-like inclusions formed by NF proteins can be found in neurons throughout the central nervous system [107,108]. An illustrative example of GFAP accumulation and aggregation is given by the Alexander disease, where mutations in this protein are linked to the formation of Rosenthal fibers (RFs) [109,110], which are dense intracellular inclusions that include a multitude of proteins ranging from cytoskeletal scaffold proteins, to intermediate filaments, to small heat shock proteins, and to ubiquitin [111]. Amyloid-like structures are formed by many proteins under appropriate conditions [112,113,114]. They are typically characterized by extremely high conformational stability, being able to sustain extremely harsh conditions [115,116,117,118,119]. 

Therefore, all five main proteins have a strong tendency to form stable and highly intertwined complexes. It is very likely that despite their typically high susceptibility to proteolysis in the unbound form, the capability of the intrinsically disordered main proteins to assemble into stable complexes “hiding” digestion sites therefore leads to their extended life time (e.g., it was already indicated that myelin proteins are among the most long-lived proteins in the body [12,13]). 

It is tempting to hypothesize that the capability of the highly disordered “main protein” to form highly stable aggregates played a role in the preservation of the Heslington brain, which is by itself represents a mystery, as an obvious problem with fossil proteins is the relatively short half-life of the peptide bond and proteolytic activity that destroys proteins after death. In fact, the half-times of the uncatalyzed hydrolysis of a single polypeptide bond via an attack by water on the peptide bond ranges from approximately 350 years for the dipeptide glycylglycine to 500 years for the C-terminal bond of acetylglycylglycine, and to 600 years for the internal peptide bond of acetylglycylglycine N-methylamide [120]. Furthermore, as was rightly pointed out by Petzold et al. [2], autolysis and decomposition caused by the massive activation of proteases and phospholipases are initiated within minutes after death [121,122]. These processes are especially rapid in the brain [122,123] and lead to the efficient elimination of protein and lipid structures [100,121]. Since, at ambient temperature, complete skeletonization typically takes 5–10 years [100], preservation of human brain tissue and brain proteins for millennia in free nature represents a very unlikely event [2,100,123]. 

These observations suggest that five main proteins from the Heslington brain serve as illustrative examples of the “stability of instability” phenomenon described for IDPs/IDRs [124], where “sturdiness of intrinsic disorder and its capability to “ignore” harsh conditions provides some interesting and important advantages to its carriers, at the molecular (e.g., the cell wall-anchored accumulation-associated protein playing a crucial role in intercellular adhesion within the biofilm of *Staphylococcus epidermidis*), supramolecular (e.g., protein complexes, biologic liquid-liquid phase transitions, and proteinaceous membrane-less organelles), and organismal levels (e.g., the case of the microscopic animals, tardigrades, or water bears, that use intrinsically disordered proteins to survive desiccation)” [124]. 

### 3.2. Global Intrinsic Disorder Analysis of the Heslington Brain Proteins

In total, the mass spectroscopy analysis of the Heslington brain samples identified 881 proteins, which, in this study, we subjected to the global intrinsic disorder analysis. Results of this analysis are summarized in Figure 6 and Figure 7. Figure 6 shows distribution of these proteins based on their PPIDR and MDS values and indicates that proteins in this dataset are characterized by noticeable levels of predicted intrinsic disorder, as evaluated by at least several of the per-residue disorder predictors used in this study.

Figure 7 indicates that the Heslington brain proteins are characterized by noticeable levels of intrinsic disorder. Figure 7A represents the results of the classification of the disorder status of these proteins based on the outputs of the per-residue disorder predictor PONDR^®^ VSL2. This classification is based the accepted field practice to group proteins based on their PPIDR values [29], where proteins with PPIDR < 10% are considered as ordered or mostly ordered, proteins with 10% ≤ PPIDR < 30% are considered as moderately disordered, and proteins with the PPIDR ≥ 30% are considered as highly disordered [29]. 

Since there is no direct correlation between the MDS and PPIDR values estimated for a query protein (e.g., a protein whose MDS ranges from 0.5 to 1.0 is predicted to be 100% disordered based on PPIDR, whereas a protein that has an MDS < 0.5 is predicted to have a PPIDR of 0%), proteins can also be classified based on their corresponding MDS values, being annotated as highly ordered (MDS < 0.15), moderately disordered or flexible (MDS between 0.15 and 0.5), and highly disordered (MDS ≥ 0.5). Based on these classification criteria, none of the Heslington brain proteins was predicted to be ordered by both MDS and PPIDR, and only 3.3% of these were predicted to be mostly ordered based on their MDS values, with remaining proteins being either moderately or highly disordered. In fact, Figure 7A shows that 42.1% of these proteins were predicted to be moderately disordered/flexible/containing noticeable IDRs based on both their MDS and PPIDR values. An additional 22.2% of the dataset was expected to be moderately disordered based on their MDS values, whereas 32.14% of the Heslington brain proteins are expected to be highly disordered, with 259 proteins (~29.3%) being predicted to have PPIDR ≥ 50% and MDS ≥ 0.5.

Combining the outputs of two binary predictors—charge-hydropathy (CH) and cumulative distribution function (CDF) plots that classify proteins as mostly ordered or mostly disordered—generates the ΔCH-ΔCDF plot that can be used for further evaluation of the global disorder status of proteins (see Section 2). Due to the principal difference between the CH and CDF plot analyses, this approach allows classification of query proteins based on their position within the CH-CDF phase space as mostly ordered, molten globule-like or hybrid, or highly disordered.

Figure 7B shows that 66.4% of the Heslington brain proteins are located within the quadrant Q1 (bottom right corner) that contains proteins predicted to be ordered by both predictors. On the other hand, 17.6% of these proteins are positioned within the quadrant Q2 (bottom left corner) that includes proteins predicted to be ordered/compact by the CH-plot and disordered by the CDF analysis (i.e., it contains either molten globular proteins, which are compact but do not have unique 3D structures, or hybrid proteins containing high levels of ordered and disordered residues). Quadrant Q3 (top left corner) includes 14.2% of the Heslington brain proteins, which are predicted to be disordered by both predictors and behave as native coils or native pre-molten globules in their unbound states. Finally, 1.8% of the Heslington brain proteins are found in quadrant Q4 (top right corner), being predicted to be disordered by CH-plot and ordered by CDF analysis. Therefore, 33.6% of the Heslington brain proteins are located outside the quadrant Q1 and can be considered as proteins with high disorder levels. 

There is a reasonably good agreement between the outputs of the tools used here for the global disorder evaluation. For example, most of the 14.2% proteins found in the quadrant Q3 of the ΔCH-ΔCDF plot can be found within the set of ~12.5% proteins that have PPIDR ≥ 50% and MDS ≥ 0.5 and are therefore predicted to be very disordered. Similarly, the “red segment” in Figure 7A contains 32.14% of the analyzed dataset, and 33.6% of the Heslington brain proteins are located outside the quadrant Q1 in Figure 7B. Also, a comparison of data shown in Figure 7 suggests that many proteins predicted to be mostly ordered by the CH-CDF analysis in fact might contain noticeable levels of disordered residues. 

To place these data for the Heslington brain proteins in perspective, Figure 8 represents the results of similar analyses of the entire human proteome (20,317 manually curated proteins). Comparison of the results shown in Figure 7A and Figure 8A suggest that the whole human proteome contains significantly (*p*-value = 2 × 10^−6^, Z-score = −4.616) more highly disordered proteins, as evidenced by the contents of the corresponding red segments, where one can find 32.4% and 39.8% of the Heslington brain and whole proteome proteins, respectively. Furthermore, 7381 human proteins (i.e., 36.3% of the whole set) are predicted to have PPIDR ≥ 50% and MDS ≥ 0.5, which is significantly (*p*-value = 0.0047, Z-score = −2.593) higher than the proportion of highly disordered proteins with PPIDR ≥ 50% and MDS ≥ 0.5 among the Heslington brain proteins (~29.3%). On the other hand, in comparison with the set of the Heslington brain proteins, the whole human proteome contains significantly fewer moderately disordered proteins (37.3% vs. 42.1%, *p*-value = 0.0024, Z-score = 2.818) and significantly more mostly ordered proteins (5.5% vs. 3.3%, *p*-value = 0.00016, Z-score = −3.602). A comparison of Figure 7B and Figure 8B shows a rather different picture, where a set of the Heslington brain proteins contains noticeably more Q3 proteins than the whole human proteome (14.2% vs. 12.3%, *p*-value = 0.056, Z-score = 1.582), significantly more Q1 proteins (66.4% vs. 59.1%, *p*-value = 4 × 10, Z-score = 4.464), and significantly fewer Q2 proteins (17.6% vs. 25.5%, *p*-value = 0, Z-score = −5.841). 

Clearly, not all 20,317 human proteins are found in the brain. Therefore, at the next stage, we analyzed 10,611 manually curated human proteins, which contain the term “brain” either in their name or in the annotation. Results of these analyses are added to Figure 8. Curiously, comparison of Figure 7A and Figure 8B showed that the Heslington brain proteins and modern brain proteins were characterized by noticeable differences in levels of intrinsic disorder. For example, the set of the Heslington brain proteins contained significantly more moderately disordered proteins than the set of modern proteins (42.1% vs. 33.9%, *p*-value = 1 × 10^−6^, Z-score = 4.688). On the other hand, highly disordered proteins were significantly more abundant among the modern brain proteins (40.1% vs. 32.4%, *p*-value = 1 × 10^−6^, Z-score = −4.735). Finally, although the modern set contained a somewhat larger number of mostly ordered proteins (4.0% vs. 3.3%), this difference was not statistically significant (*p*-value = 0.133, Z-score = −1.111). 

In comparison with the 10,611 brain-related proteins from the whole human proteome, the Heslington brain proteins show a 1.27- and 1.13-times increase in the content of quadrants Q3 and Q1, respectively and a 1.57-times decrease in the proteins populating quadrant Q2 (cf. Figure 7B and Figure 8B). In other words, the content of the Heslington brain proteins in the quadrant Q1 was significantly higher than that of modern brain proteins (66.4% vs. 58.1%, *p*-value = 0, Z-score = 4.947); quadrant Q2 contained significantly fewer Heslington brain proteins (17.6% vs. 27.7%, *p*-value = 0, Z-score = −7.710), quadrant Q3 included significantly more Heslington brain proteins (14.2% vs. 11.2%, *p*-value = 0.0072, Z-score = 2.444), and quadrant Q4 contained significantly fewer Heslington brain proteins than modern brain proteins (1.8% vs. 2.6%, *p*-value = 0.043. Z-score = −1.711). 

### 3.3. Functional Intrinsic Disorder of Proteins Potentially Interacting with Main Proteins

As explained in the Section 2, interactors for the five main proteins (NFH, NFM, NFL, GFAP, and MBP) were determined using APID. A list of the interactors that were also detected in the preserved Heslington brain using mass spectroscopy was then made. This list included 44 interactors with at least one of the five main proteins. Results of the global intrinsic disorder analysis in these interactors are summarized in Figure 7. Figure 7A shows that 20 interactors (45.5%) are located within the red area, indicating that they are predicted to be highly disordered proteins. This content significantly exceeded 32.4% of highly disordered proteins found in the Heslington brain (*p*-value = 0.026; Z-score = −1.929). The remaining 25 proteins are moderately disordered, being spread between the dark pink and light pink areas that contain 14 (31.8%) and 10 (22.7%) proteins, respectively. Therefore, the level of moderately disordered proteins in the Heslington brain (64.3%) was somewhat larger than that of the interactors (54.5%) (*p*-value = 0.109; Z-score = 1.230). Since blue and cyan areas in Figure 7A did not have any points corresponding to the interactors, none of these proteins was predicted to be mostly ordered. As per CH-CDF analysis (Figure 7B), interactors are found in quadrants Q1 (22 proteins, 50%), Q2 (16 proteins, 36.4%), and Q3 (six proteins, 13.6%), supporting the idea that, in general, the set of interactors is noticeably more disordered than the proteins identified in the preserved Heslington brain (see Figure 7): Q1 (66.4% vs. 50.0%, *p*-value = 0.034; Z-score = 1.824), Q2 (17.6% vs. 36.4%, *p*-value = 0.0001; Z-score = −3.718), and Q3 (14.2% vs. 13.6%, *p*-value = 0.433; Z-score = 0.169), or the whole human proteome (Q1 (59.1% vs. 50.0%, *p*-value = 0.114; Z-score = −1.206), Q2 (25.5% vs. 36.4%, *p*-value = 0.067; Z-score = 1.501), and Q3 (12.3% vs. 13.6%, *p*-value = 0.401; Z-score = 0.251)) and human brain proteins (see Figure 8) (58.5% vs. 50.0%, *p*-value = 0.131; Z-score = −1.125), Q2 (27.2% vs. 36.4%, *p*-value = 0.103; Z-score = 1.266), and Q3 (11.2% vs. 13.6%, *p*-value = 0.321; Z-score = 0.464)). 

Figure 9A represents the STRING-generated PPI network between the five main proteins and their 44 interactors. This network was generated using medium confidence level of 0.4 to insure maximal inclusion of interactors. The resulting network contains 48 proteins connected via 163 interactions. 

This network is characterized by an average node degree of 6.79 and an average local clustering coefficient of 0.489. Since the expected number of interactions for a random set of proteins of the same size and degree distribution drawn from the genome is 44, this network has significantly more interactions than expected (*p*-value < 10^−16^). Figure 9B shows PPI network centered at these proteins. Here, the network includes 548 proteins linked by 15,492 interactions. 

The PPI network shown in Figure 9B includes significantly (*p*-value < 10^−16^) more interactions than expected (8171) and is characterized by an average node degree of 56.5 and an average local clustering coefficient of 0.54. Proteins included in this network preferentially participate in the following biological processes: Negative regulation of biological process (GO:0048519; *p* = 7.02 × 10^−79^), Symbiotic process (GO:0044403; *p* = 3.76 × 10^−76^),Viral process (GO:0016032; *p* = 3.75 × 10^−76^), Regulation of metabolic process (GO:0019222; *p* = 4.05 × 10^−74^), and Response to organic substance (GO:0010033; *p* = 2.26 × 10^−73^). The most enriched molecular functions of these proteins are Protein binding (GO:0005515; *p* = 2.33 × 10^−60^), Enzyme binding (GO:0019899; *p* = 1.71 × 10^−50^), Binding (GO:0005488; *p* = 1.09 × 10^−39^), RNA binding (GO:0003723; *p* = 1.74 × 10^−38^), and Beta-catenin binding (GO:0008013; *p* = 5.07 × 10^−38^), and they serve as cellular components in Protein-containing complex (GO:0032991; *p* = 2.97 × 10^−95^), Cytosol (GO:0005829; *p* = 8.38 × 10^−88^), Vesicle (GO:0031982; *p* = 4.93 × 10^−63^), Intracellular (organelle GO:0043229; *p* = 5.81 × 10^−61^), and Extracellular exosome (GO:0070062; *p* = 6.55 × 10^−61^).

Uploading these main proteins and their interactors to DAVID and the use of the KEGG pathway database revealed that out of the 49 that were uploaded, only 39 had hits in the KEGG pathway database, and only the NFs had pathway hits out of the five main proteins.

Table 1 summarizes these results and shows that six pathways that involve the formation of aggregates appear to be associated with the interactors, with some of those also including NFs. These are neurodegeneration, Alzheimer’s disease, ALS, fluid shear stress atherosclerosis, Parkinson’s disease, and lipid atherosclerosis pathways.

We are reporting here the results of the comprehensive bioinformatics analysis of the prevalence of functional intrinsic disorder in five proteins (neurofilament proteins (neurofilament heavy (NFH), neurofilament middle (NFM), and neurofilament light (NFL)), glial fibrillary acidic protein (GFAP), and myelin basic protein (MBP)) that were established as playing an important role in the preservation of the Heslington brain (which is at least 2600 years old). This analysis revealed that these “main” proteins are extensively disordered and that four of them (NFH, NFM, NFL, and GFAP) are likely to undergo noticeable folding induced by the assembly of the neurofilaments. On the other hand, MBP, which is crucial for the maintaining the structural integrity of the myelin sheath, is found “in the major dense lines, electron-dense lamellae formed by the tight apposition of the cytoplasmic leaflets of the oligodendrocyte membrane” [73]. The ability of MBP to cause tight adhesion of the two lipid bilayers and also to act as a scaffold protein that binds many other proteins to the membrane is determined by the peculiarities of its amino acid sequence, where there are no specific membrane-binding domains and the basic residues distributed over its entire length rather than in a cluster [73]. Therefore, it seems that these disorder-based functional features of the main proteins allow them to serve as long-lasting preservatives via formation of tightly packed intramolecular complexes. It is likely that the formation of these highly intertwined and densely packed assemblies makes these proteins highly resistant to degradation and helps them and their partner to be sustained for a long time (at least 2600 years). 

Comparison of these five proteins with the remaining proteins from the Heslington brain (as well as the whole human proteome or human brain proteins) revealed that NFH, NFM, NFL, GFAP, and MBP are positioned among the top 10% of the most disordered proteins in these datasets. Furthermore, we also established that many of the proteins known to interact with NFH, NFM, NFL, GFAP, and MBP are predicted to be highly disordered as well. Our analysis also revealed that the Heslington brain, which was preserved for more than 2600 years, contains surprisingly high levels of highly disordered proteins. This finding is rather counterintuitive, as due to the well-known high accessibility of IDPs and IDRs to the proteolytic cleavage [17,125,126,127,128,129], one would expect almost complete elimination of the proteins with the noticeable disorder levels from the Heslington brain. The only reasonable explanation for these observations is the capability of IDRs/IDPs to act as a molecular mortar or cement that glue together various brain proteins and rigidify the resulting assemblies, thereby generating highly stable matter. In other words, high disorder content in the Heslington brain proteins serves as another illustration of one of the disorder-related paradoxes of protein universe, namely the stability of instability discussed earlier. 

Curiously, although preservation of the brain tissue for more than 2600 years seems to be a miracle, the Heslington brain is not the only ancient human brain tissue uncovered by archeologists. Excavations at Kanaljorden in Sweden uncovered ~8000-year-old brain material inside human skulls that had received an underwater burial [130]. Although no proteomic analysis has been conducted on the samples of the Kanaljorden brain as of yet, it is tempting to hypothesize that the noticeable fraction of the Kanaljorden brain proteins would have high levels of predicted intrinsic disorder as well. Future research is needed to verify this exciting hypothesis. 

Finally, much older proteins were recently identified in fossil coral specimens with the youngest ages of 125 to 138 kiloanna (ka) [131]. The well-preserved samples of Pleistocene *Orbicella annularis* contained a set of six ancient coral skeletal proteins [131]. However, there is an obvious and dramatic difference between the proteins analyzed in our study and proteins identified in [131]. In fact, we considered proteins from the preserved brain tissues, whereas the ancient coral skeletal proteins analyzed by Drake et al. were specialized extracellular proteins associated with the biomineralization process that became protected from degradation for long periods of time due to the embedding into the individual skeleton crystals [131].

## 4. Conclusions

Based on the results of the bioinformatics analyses of the previously identified [2] Heslington brain proteins, some important conclusions can be reached on the potential roles of intrinsic disorder in ancient brain samples.

Five main proteins that are assumed to be responsible for the preservation of the Heslington brain (which is at least 2600 years old), NFH, NFM, NFL, GFAP, and MBP are predicted to be highly disordered.44 proteins from the Heslington brain, which are expected to interact with these five main proteins, are also predicted to have high levels of intrinsic disorder.Contrarily to the expected substantial (if not complete) elimination of the disordered proteins from the brain found inside a skull buried in a pit in Heslington, Yorkshire, England, many proteins in this Heslington brain are predicted to be highly disordered, with most of these proteins being expected to contain noticeable levels of intrinsic disorder.Intrinsic disorder of NFH, NFM, NFL, GFAP, and MBP and their interactors (in combination with other factors, such as the way in which the person was buried) might play a crucial role in preserving the Heslington brain by forming tightly folded brain protein aggregates.

## Figures and Tables

**Figure 6 biology-11-01704-f006:**
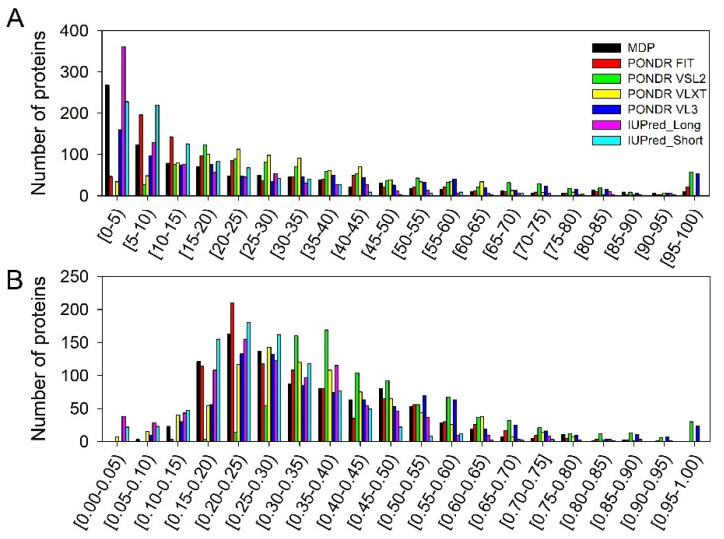
Distribution of the Heslington brain proteins based on their PPIDR (**A**) and MDS (**B**) values evaluated by various per-residue disorder predictors utilized in this study.

**Figure 7 biology-11-01704-f007:**
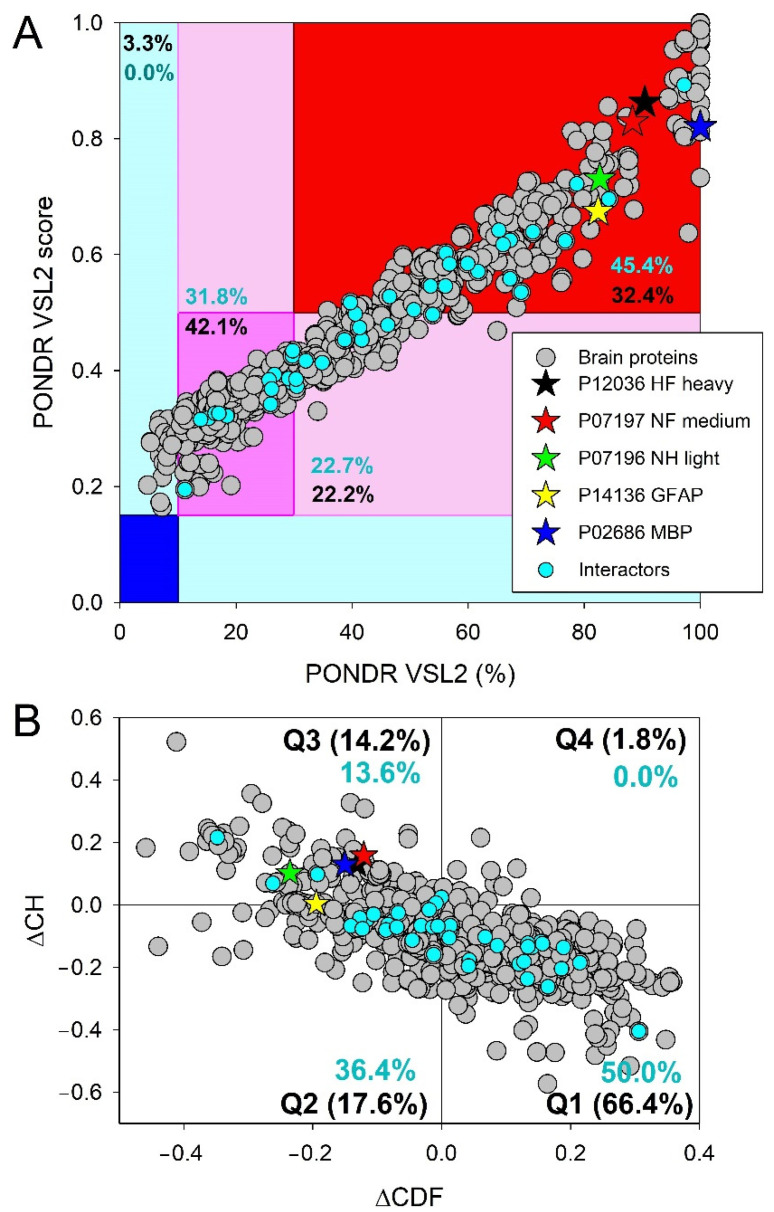
Evaluation of the global disorder status of 881 human proteins found in the Heslington brain samples (gray circles). Corresponding data for five main proteins are shown by colored stars, whereas data for their interactors are shown by small cyan circles. (**A**) PONDR^®^ VSL2 PONDR^®^ VSL2 score vs. PONDR^®^ VSL2 (%) plot. Here, each point corresponds to a query protein, the coordinates of which are evaluated from the corresponding PONDR^®^ VSL2 data as its mean disorder score (MDS) and percent of the predicted intrinsically disordered residues (PPIDR). Color blocks are used to visualize proteins based on the accepted classification, with red, pink/light pink, and blue/light blue regions containing highly disordered, moderately disordered, and ordered proteins, respectively (see the text). Dark blue or pink regions correspond to the regions, where PPIDR agrees with MDS, whereas areas in which only one of these criteria applies are shown by light blue or light pink. (**B**) CH-CDF plot, where coordinates for a protein are calculated as the average distance of its CDF curve from the CDF boundary (X axis) and its distance from the CH boundary. Protein classification is based on the quadrant, where it is located: Q1, protein predicted to be ordered by both predictors. Q2, protein predicted to be ordered by CH-plot and disordered by CDF. Q3, protein predicted to be disordered by both predictors. Q4, protein predicted to be disordered by CH-plot and ordered by CDF.

**Figure 8 biology-11-01704-f008:**
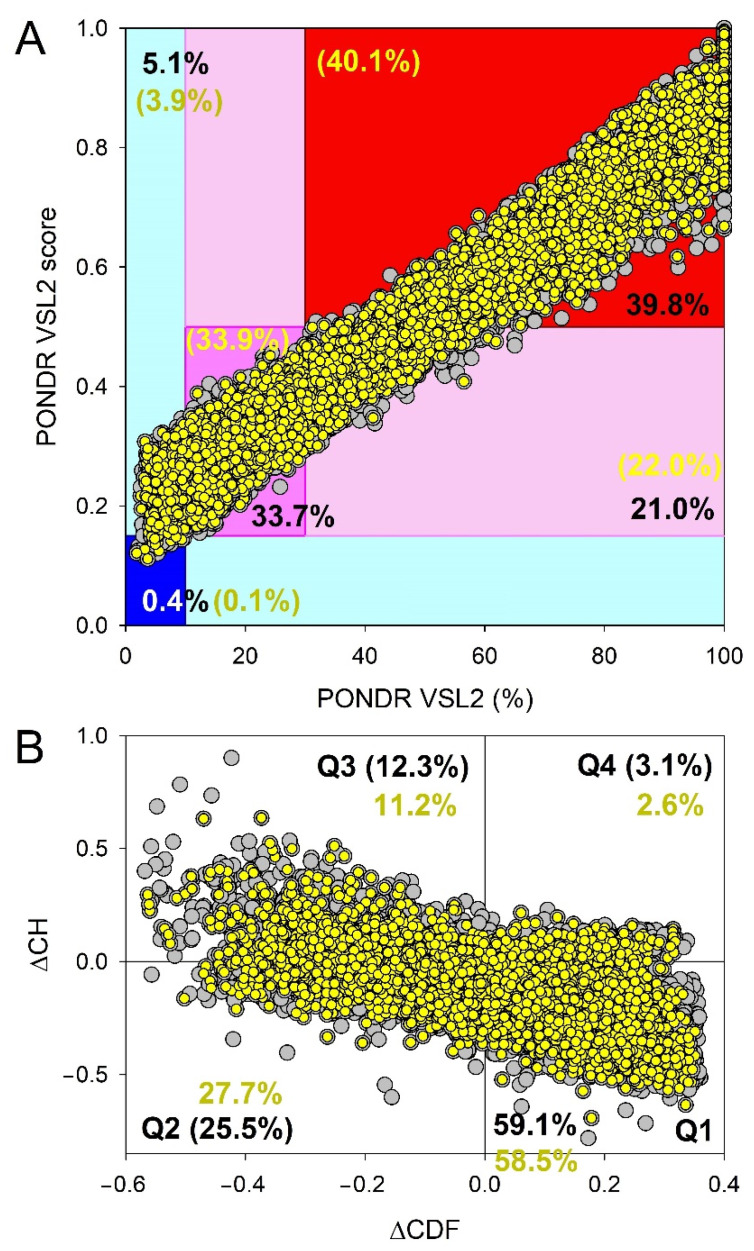
Evaluation of the global disorder status of 20,317 human proteins (gray circles) and 10,611 brain proteins (small yellow circles). (**A**) PONDR^®^ VSL2 output. (**B**) Charge-hydropathy and cumulative distribution function (CH-CDF) plot. For explanations, see legend to Figure 7.

**Figure 9 biology-11-01704-f009:**
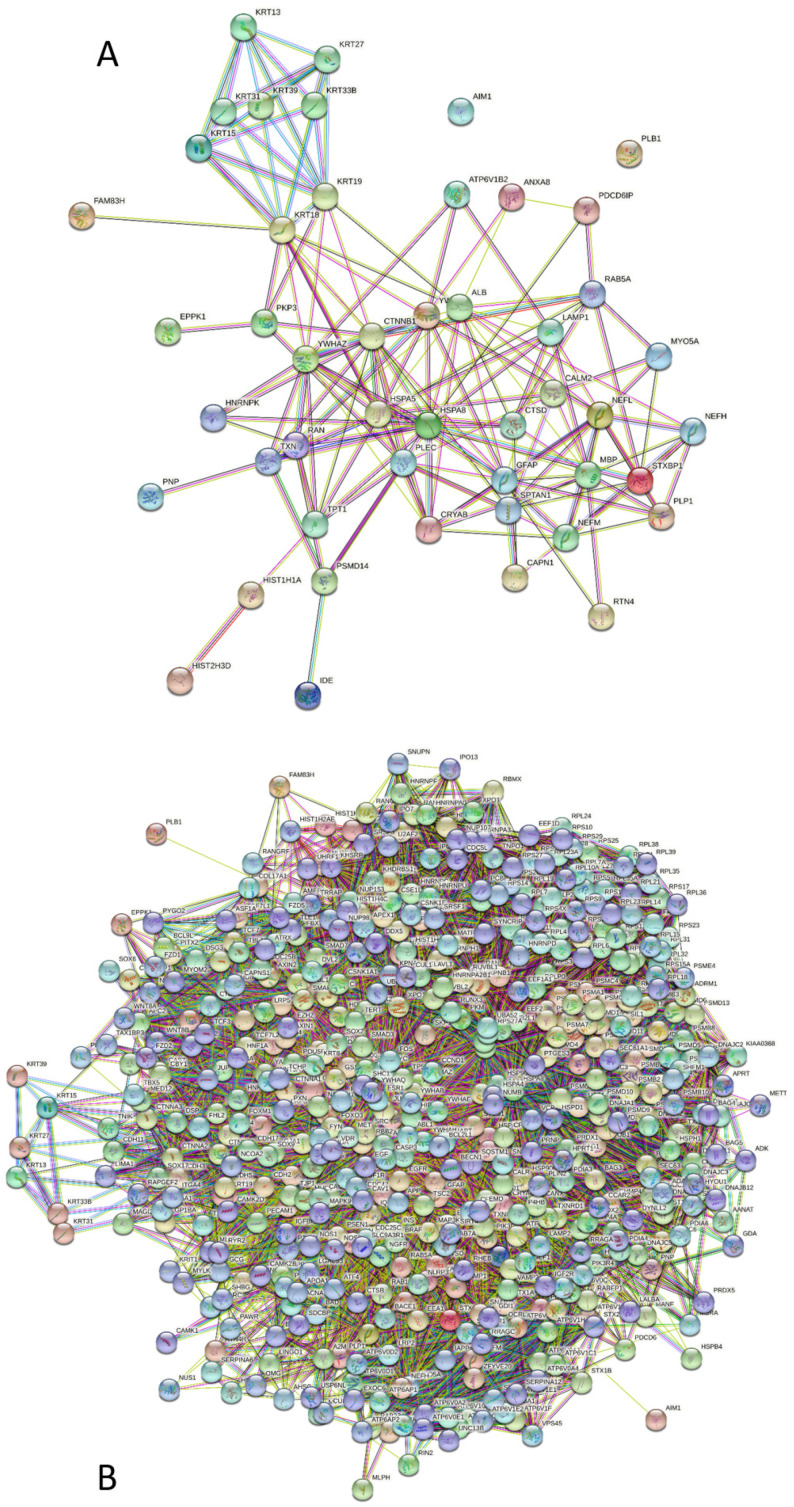
STRING-generated PPI network between NFs, GFAP, MBP and their 44 interactors found in the Heslington brain sample using APID (**A**), and a network that also includes a 1st shell of interactors of these proteins (**B**).

**Table 1 biology-11-01704-t001:** Potential pathways, according to the KEGG database, involving interactors detected in the brain and the main proteins. Forty-nine proteins were uploaded to DAVID, and only 39 had KEGG pathway hits.

Term	Genes	*p*-Value	Count
hsa04915:Estrogen signaling pathway	KRT33B, HSPA8, KRT19, KRT18, KRT39, KRT27, KRT15, KRT13, CALM1, KRT31, CALM2, CTSD	1.78 × 10^−11^	12
hsa05150:Staphylococcus aureus infection	KRT33B, KRT19, KRT18, KRT39, KRT27, KRT15, KRT13, KRT31	2.35 × 10^−7^	8
hsa05022:Pathways of neurodegeneration—multiple diseases	PSMD14, HSPA5, NEFL, CTNNB1, NEFM, CAPN1, CALM1, CALM2, NEFH, RAB5A	2.59 × 10^−4^	10
hsa05010:Alzheimer disease	PSMD14, CTNNB1, CAPN1, IDE, CALM1, CALM2, RTN4	0.00804	7
hsa05152:Tuberculosis	LAMP1, CALM1, CALM2, CTSD, RAB5A	0.00943	5
hsa04114:Oocyte meiosis	CALM1, CALM2, YWHAZ, YWHAG	0.02269	4
hsa05014:Amyotrophic lateral sclerosis	PSMD14, HSPA5, NEFL, NEFM, NEFH, RAB5A	0.02575	6
hsa05418:Fluid shear stress and atherosclerosis	CTNNB1, TXN, CALM1, CALM2	0.02646	4
hsa05012:Parkinson disease	PSMD14, HSPA5, TXN, CALM1, CALM2	0.03424	5
hsa04141:Protein processing in endoplasmic reticulum	HSPA8, HSPA5, CAPN1, CRYAB	0.04472	4
hsa05417:Lipid and atherosclerosis	HSPA8, HSPA5, CALM1, CALM2	0.07770	4
hsa04916:Melanogenesis	CTNNB1, CALM1, CALM2	0.08006	3

## Data Availability

Data are contained within the article or Appendix A.

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
