# Peer review of "Intrinsic Disorder as a Natural Preservative: High Levels of Intrinsic Disorder in Proteins Found in the 2600-Year-Old Human Brain"

_biology, 2022, doi:10.3390/biology11121704_

Round 1

Reviewer 1 Report

The authors are following up on some prior work (references 1 and 2) wherein intact proteins were found in human brain tissue. Based on five "main" proteins which were IDPs, the authors have done a bioinformatic analysis to show that these proteins tend to interact with other disordered proteins, and therefore have concluded (or extended the conclusion of reference 1 and 2) that disordered proteins played a key role in preserving this brain tissue.

I feel the manuscript would benefit from these changes:

(1) The identification of these 5 proteins as "main" proteins should be clarified. Perhaps this was in the references 1 and 2, but since so much of this paper hinges on these "main" proteins being the key, the selection of these proteins as "main" should be made very clear here.

(2) disordered proteins are not uncommon. Also, disordered proteins interacting with other IDPs is not uncommon. In order for the interactions to be important/noteworthy, the statistics of how many proteins by mass or by number are IDPs in the "old" tissue should be compared with the same analysis of "new" tissue. Perhaps this is discussed at some point, but it needs to be front and center, very clear, in the summary, abstract and conclusions. I don't see it in any of those places.

(3) In the absence of this sort of statistical justification, the presence of IDPs and their interactions is not a compelling argument for their importance for preservation.

A few minor comments:

line 21 has repetition

line 33 has language problems

please clarify if lines 66-83 represents work from references 1-2

line 98-99: some IDPs do fold under certain conditions, including upon binding to a target

Fig 1A:  should say dashed "horizontal" lines

Line 737-739: Furthermore used twice

Line 754: I don't understand what is meant by ....

Author Response

Reviewer #1

The authors are following up on some prior work (references 1 and 2) wherein intact proteins were found in human brain tissue. Based on five "main" proteins which were IDPs, the authors have done a bioinformatic analysis to show that these proteins tend to interact with other disordered proteins, and therefore have concluded (or extended the conclusion of reference 1 and 2) that disordered proteins played a key role in preserving this brain tissue.

RESPONSE: We are thankful to this reviewer for careful reading of our manuscript and for providing constructive critiques. We tried to address all the issues pointed by the reviewer and amended manuscript accordingly.

I feel the manuscript would benefit from these changes:

(1) The identification of these 5 proteins as "main" proteins should be clarified. Perhaps this was in the references 1 and 2, but since so much of this paper hinges on these "main" proteins being the key, the selection of these proteins as "main" should be made very clear here.

RESPONSE. Thank you for pointing this out. The corresponding clarification was added to the revised manuscript. We also added a section providing brief description of these five main proteins.

(2) disordered proteins are not uncommon. Also, disordered proteins interacting with other IDPs is not uncommon. In order for the interactions to be important/noteworthy, the statistics of how many proteins by mass or by number are IDPs in the "old" tissue should be compared with the same analysis of "new" tissue. Perhaps this is discussed at some point, but it needs to be front and center, very clear, in the summary, abstract and conclusions. I don't see it in any of those places.

RESPONSE: Thank you for pointing this out. We included the corresponding data and their discussion to the revised manuscript (pages 18-20).

(3) In the absence of this sort of statistical justification, the presence of IDPs and their interactions is not a compelling argument for their importance for preservation.

RESPONSE: We included statistical analysis to the revised manuscript. However, we also would like to emphasize that the presence of very high proportion of moderately disordered proteins in the Heslington brain samples is very counterintuitive, as disordered proteins and regions are known to serve as primary targets for various proteases. As a result, many IDPs are characterized by shorter lifetimes than ordered proteins. The situation is different when IDRs or IDPs form complexes, many of which are highly intertwined and therefore very stable. As we indicated in the manuscript, it seems that the intrinsic disorder of NFH, NFM, NFL, GFAP, and MBP, their interactors and many other proteins might play a crucial role in preserving the Heslington brain by forming tightly folded brain protein aggregates, in which different parts are glued together via the disorder-to-order transitions. We further elaborated this subject and added some discussion of the related “stability of instability” phenomenon associated with intrinsic disorder.

A few minor comments:

line 21 has repetition

RESPONSE: fixed

line 33 has language problems

RESPONSE: fixed

please clarify if lines 66-83 represents work from references 1-2

RESPONSE: fixed

line 98-99: some IDPs do fold under certain conditions, including upon binding to a target

RESPONSE: Thank you for pointing this out. The corresponding clarification is added.

Fig 1A:  should say dashed "horizontal" lines

RESPONSE: Fixed

Line 737-739: Furthermore used twice

RESPONSE: Fixed

Line 754: I don't understand what is meant by ....

RESPONSE: Fixed

Reviewer 2 Report

The comments are available in the attached file. 

Author Response

Reviewer #2

The article titled “Intrinsic disorder as a natural preservative: High levels of intrinsic disorder in proteins found in the 2,600-year-old human brain” described the preservations of proteins in the excavated human brain were rich in intrinsically disordered regions (IDRs). Focused on the five core members of the proteins, the authors extensively analyzed the protein-protein interaction networks among the proteins found in the preservative, which were also rich in IDRs.

The comprehensive use of various bioinformatics revealed that the protein-protein interactions among the proteins harboring IDRs should form highly stable protein assemblies that can survive over 2,600 years. The authors clearly and technically soundly described the notable features of the intrinsically disordered proteins (IDPs) that can allow them for surviving long-term in buried bodies.

The manuscript was very carefully prepared with extensive bioinformatics evidence. I recommend accepting this article in its present form.

RESPONSE: We are thankful to this reviewer for high evaluation of our work.

Reviewer 3 Report

The extremely exciting title of the manuscript suggests the protective nature of the intrinsic disorder of selected brain proteins, which have been preserved for 2,600 years. An obvious problem with fossil proteins is the relatively short half-life of the peptide bond (350-600 years) and proteolytic activity that destroys proteins after death. The authors used a range of bioinformatics tools to analyze the structure-function relationships of 5 "major" proteins discovered  with MS/MS proteomic studies in the fossil "Heslington brain". It turned out that these proteins, as well as their 44 partners with which they potentially interact, show more than a statistical portion of unstructured regions. Interestingly, 4 of them have long alpha-helical fragments. Do the authors have any ideas on defining the role of these fragments for protein interactions/function ?  The Alphafold models of structures do not always correspond to NMR or cryo-EM structures, however, the authors' conclusions are interesting, important and very inspiring.  

Of course, demonstrating the specificity of the interactions requires experiments. First, it would be interesting to prove experimentally the protective effect, by showing that the indicated IDPs, despite the typical susceptibility to proteolysis, form amyloid structures "hiding" digestion sites. Perhaps in the discussion, the authors would suggest specific experiments. Similarly, the sentence "main proteins …. are likely to undergo noticeable folding induced by the assembly of the neurofilaments" (page 21 under Tab. 1) seems over the top and requires experimental confirmation.  

The sentence about the "miraculous" preservation of proteins from 2,600 years ago (the end of the manuscript) may not be entirely true, after all, much older proteins manage to be found in fossil corals (Drake, J.L. et al. 2020. First sequencing of ancient coral skeletal proteins. Scientific Reports. 10(1): 1-11.).

The publication is not easy to read because of the multitude of predictors used. I would ask the authors to explain the significance of the differences in p-coefficient, whose values are e.g. between 10-39 and 10-79  (the comments to Fig. 9).

Author Response

Reviewer #3

The extremely exciting title of the manuscript suggests the protective nature of the intrinsic disorder of selected brain proteins, which have been preserved for 2,600 years. An obvious problem with fossil proteins is the relatively short half-life of the peptide bond (350-600 years) and proteolytic activity that destroys proteins after death. The authors used a range of bioinformatics tools to analyze the structure-function relationships of 5 "major" proteins discovered with MS/MS proteomic studies in the fossil "Heslington brain". It turned out that these proteins, as well as their 44 partners with which they potentially interact, show more than a statistical portion of unstructured regions.

RESPONSE: We are thankful to this reviewer for careful reading of our manuscript and for providing constructive critiques. We tried to address all the issues pointed by the reviewer and amended manuscript accordingly.

Interestingly, 4 of them have long alpha-helical fragments. Do the authors have any ideas on defining the role of these fragments for protein interactions/function ?  The Alphafold models of structures do not always correspond to NMR or cryo-EM structures, however, the authors' conclusions are interesting, important and very inspiring.  

RESPONSE: Thank you for pointing this out. As a matter of fact, long helical segments found in these proteins correspond to their rod domains, which are engaged in the formation highly intertwined coiled-coil dimers. Clearly, these long α-helical segments cannot exist in isolation, as there are no physical force(s) that would be able to stabilize such a structure in aqueous media. However, these long α-helical segments are easily formed and stabilized at the coiled-coil formation. They clearly represents an extreme case of the folding-induced binding phenomenon described for many IDPs. Furthermore, coiled-coils are often cross-predicted as unstructured regions [PMID: 20303956; 22009114]. Therefore, the structural models generated for long coiled-coil segments of these (and many other) proteins by AlphaFold reflects the potential of this AI platform (which is a machine learning approach incorporating multi-sequence alignments, as well as physical and biological information on protein structure into a deep learning algorithm [PMID: 34265844]) to find such disordered regions with high folding-upon-binding potential. In agreement with these considerations, a recent report indicated that AlphaFold2 can capture the folded state structure of IDRs known to fold under specific conditions (Systematic identification of conditionally folded intrinsically disordered regions by AlphaFold2 T. Reid Alderson, Iva Pritišanac, Alan M. Moses, Julie D. Forman-Kay bioRxiv 2022.02.18.481080). In fact, the authors of this preprint indicated: “AlphaFold2 assigns high-confidence scores to about 60% of a set of 350 IDRs that have been reported to conditionally fold, suggesting that AlphaFold2 has learned to identify conditionally folded IDRs, which is unexpected, since IDRs were minimally represented in the training data”. The corresponding discussion is added to the revised manuscript.

Of course, demonstrating the specificity of the interactions requires experiments. First, it would be interesting to prove experimentally the protective effect, by showing that the indicated IDPs, despite the typical susceptibility to proteolysis, form amyloid structures "hiding" digestion sites. Perhaps in the discussion, the authors would suggest specific experiments. Similarly, the sentence "main proteins …. are likely to undergo noticeable folding induced by the assembly of the neurofilaments" (page 21 under Tab. 1) seems over the top and requires experimental confirmation.  

RESPONSE: Thank you for pointing this out. Discussion of these issues (together with several other aspects) is included now to a new section “3.1.6. The “main proteins” as illustrations of the “stability of instability” phenomenon”

The sentence about the "miraculous" preservation of proteins from 2,600 years ago (the end of the manuscript) may not be entirely true, after all, much older proteins manage to be found in fossil corals (Drake, J.L. et al. 2020. First sequencing of ancient coral skeletal proteins. Scientific Reports. 10(1): 1-11.).

RESPONSE: Although there is a dramatic difference between the brain proteins considered in our study and coral skeletal proteins discussed in the mentioned article, we added a brief discussion of this interesting work.

The publication is not easy to read because of the multitude of predictors used. I would ask the authors to explain the significance of the differences in p-coefficient, whose values are e.g. between 10-39 and 10-79  (the comments to Fig. 9).

RESPONSE: Thank you for pointing this out. Utilization of multiple computational tools is an accepted practice in the field. Therefore, we were following in this study the established protocol. Reported p-values are the reflection the statistical significance of the observations. This parameter is a commonly used statistical measurement used to validate a hypothesis against observed data, with p-values below 0.05 being typically used as an indication of the statistically significant observation. Therefore, both p-values of 10-39 and 10-79  are related to the significantly enriched functions/processes/cellular components. 

Round 2

Reviewer 1 Report

The minor changes requested have been made.

I don't see improvement on these two questions:

(1) what is the significance/justification for calling these five proteins "main". Still just says the previous author called them that and said they are important.

(2) statistical analysis, if I interpret correctly, does NOT show that this very old brain tissue has higher IDP content than the normal genome. So I don't see that the point is made that these IDPs are responsible for the preservation.

Author Response

Reviewer #1:

The minor changes requested have been made.

RESPONSE: We respectfully disagree with this statement. As a matter of fact, all issues pointed out by this reviewer were addressed during resubmission. As a matter fact, the manuscript underwent major rewrite to address issues indicated by this reviewer in the first round.

I don't see improvement on these two questions:

(1) what is the significance/justification for calling these five proteins "main". Still just says the previous author called them that and said they are important.

RESPONSE: In the first review, this reviewer indicated: “The identification of these 5 proteins as "main" proteins should be clarified. Perhaps this was in the references 1 and 2, but since so much of this paper hinges on these "main" proteins being the key, the selection of these proteins as "main" should be made very clear here.”

This critique was addressed by the following response: “Thank you for pointing this out. The corresponding clarification was added to the revised manuscript. We also added a section providing brief description of these five main proteins.” The following text (highlighted in yellow) was added to the Introduction of section of the R1 revised manuscript:

Among identified proteins with this extraordinary long-term stability and ability to survive for 2,600 years were proteins engaged in the formation non-amyloid protein aggregates, such as neurofilament (NF) proteins (or NFs), glial fibrillary acidic protein (GFAP), and myelin basic protein (MBP) [2]. Based on these observations, the authors concluded that the preservation of brain for millennia was driven by the formation of protein aggregates, in which NF proteins, GFAP, and MBP played crucial roles [2]. The authors also pointed out that these proteins were strongly integrated into the aggregated mass, as, for example, it took 2-12 months to detect the resolution of intermediate filaments (IFs) from the protein aggregates [2]. Due to the likely important roles of these proteins (NF proteins, GFAP, and MBP) in the long-term preservation of the Heslington brain, they were defined as the “main proteins” in our study. Curiously, our analysis revealed that all these “main proteins” found in the Heslington brain are characterized by the high levels of intrinsic disorder.

The analysis of the micro-structure of the Heslington brain first began with immunoelectron microscopy in order to determine if there was a presence of specific axonal proteins [1,2]. Next, sensitive immunoassays were carried out to screen for other brain proteins that may have been present [2]. There were strong immune responses that indicated the presence of GFAP and MBP and weak immune responses that indicated the presence of NFs.

NF heavy (NFH), an NF subunit, was found to be present in the axons of the ancient tissue. The detection of an NF subunit indicated that other NF subunits, neurofilament middle (NFM) and neurofilament light (NFL), should be present as well. NFs are type IV intermediate filaments (IFs), which are cytoskeletal fibers found in eukaryotic neurons. Neurofilaments are necessary for the radial growth and increase of the diameter of axons [3].

GFAP is a type III IF that provides structural stability to astrocytic processes which results in the modulation of astrocyte motility and shape [4]. Being members of the IF protein family (GFAP is a type III and NFs are type IV intermediate filaments), these proteins have similar domain organization with a conserved coiled-coil central domain (which is due to the rod-like tertiary structure within the IFs is known as rod domain) flanked by variable N-terminal “head” and C-terminal “tail” domains (or side-arms) and play crucial role in the biogenesis of the neuronal cytoskeleton. To form neuronal cytoskeleton, these structural proteins undergo complex, multistage assembly process that starts with the formation of α-helical coiled-coil dimers, where the central rod domains coil around each other. These dimers then assemble side-by-side to form the basic subunits of IFs, staggered tetramers [5-7]. On the other hand, non-helical (and highly disordered, see below) head and tail domains are needed for the IF formation, as rod domains alone do not form filaments [8].

MBP is a part of the insulating myelin sheath, a multilayered proteolipid membrane surrounding nerve axons and is the second most abundant protein in the central nervous system [9]. Although MBP is a cytosolic protein, it constitutes 30% of the total myelin, being a major element of the compact myelin, in which cytoplasm is excluded, and where the cytoplasmic leaflets of myelin membrane forming the consecutive turns are practically fused, resulting in the extracellular space, which is only about 2 nm thick [10]. MBP is one of the myelin proteins that hold together a repetitive multilayer of tightly packed lipid bilayers [11]. The overall importance of myelin, this macroscopic supramolecular proteolipid structure, for the nervous system functionality is reflected in the fact that myelin proteins are among the most long-lived proteins in the body [12,13].

Furthermore, a new section 3.1.6. was added to the R1 version to further explain importance of “main” proteins and their intrinsic disorder nature:  

3.1.6. The “main proteins” as illustrations of the “stability of instability” phenomenon

It is known that in addition to be responsible for the formation of cellular scaffolds of neurons, under some pathological conditions, IF proteins can self-assemble into intra- and extracellular aggregates [3,100-106]. For example, Lewy body-like inclusions formed by NF proteins can be found in neurons throughout the central nervous system [107,108]. An illustrative example of GFAP accumulation and aggregation is given by the Alexander disease, where mutations in this protein are linked to the formation of Rosenthal fibers (RFs) [109,110], which are dense intracellular inclusions that include a multitude of proteins ranging from cytoskeletal scaffold proteins, to intermediate filaments, to small heat shock proteins, and to ubiquitin [111]. Amyloid-like structures are formed by many proteins under appropriate conditions [112-114]. They are typically characterized by extremely high conformational stability, being able to sustain extremely harsh conditions [115-119].

Therefore, all five “main proteins” have a strong tendency to form stable and highly intertwined complexes. It is very likely that despite their typically high susceptibility to proteolysis in the unbound form, the capability of the intrinsically disordered “main proteins” to assemble into stable complexes "hiding" digestion sites and therefore leading to their extended life time (e.g., it was already indicated that myelin proteins are among the most long-lived proteins in the body [12,13]).

It is tempting to hypothesize that the capability of the highly disordered “main protein” to form highly stable aggregates played a role in the preservation of the Heslington brain, which is by itself represents a mystery, as an obvious problem with fossil proteins is the relatively short half-life of the peptide bond and proteolytic activity that destroys proteins after death. In fact, the half-times of the uncatalyzed hydrolysis of a single polypeptide bond via the attack by water on the peptide bond ranges from approximately 350 years for the dipeptide glycylglycine to 500 years for the C-terminal bond of acetylglycylglycine, and to 600 years for the internal peptide bond of acetylglycylglycine N-methylamide [120]. Furthermore, as was rightly pointed by Petzold et al. [2], autolysis and decomposition caused by the massive activation of proteases and phospholipases are initiated within minutes after the death [121,122]. These processes are especially rapid in brain [122,123] and lead to the efficient elimination of protein and lipid structure [100,121]. Since at ambient temperature, complete skeletonization typically takes 5–10 years [100], preservation of human brain tissue and brain proteins for millennia in free nature represents a very unlikely event [2,100,123].

These observations suggest that five “main proteins” from the Heslington brain serve as illustrative examples of the “stability of instability” phenomenon described for IDPs/IDRs [124], where “sturdiness of intrinsic disorder and its capability to "ignore" harsh conditions provides some interesting and important advantages to its carriers, at the molecular (e.g., the cell wall-anchored accumulation-associated protein playing a crucial role in intercellular adhesion within the biofilm of Staphylococcus epidermidis), supramolecular (e.g., protein complexes, biologic liquid-liquid phase transitions, and proteinaceous membrane-less organelles), and organismal levels (e.g., the case of the microscopic animals, tardigrades, or water bears, that use intrinsically disordered proteins to survive desiccation)” [124].

Furthermore, the first paragraph of Results section  of the original manuscript contained the following statement (which is also present in the revised manuscript): At the first stage of our study, we looked at the intrinsic disorder predisposition of five “main proteins”, the neurofilament heavy, medium, and light chain proteins (NFH, NFM, and NFL, UniProt IDs: P12036, P07197, P07196, respectively), glial fibril-lary acidic protein (GFAP; UniProt ID: P14136), and myelin basic protein (MBP; Uni-Prot ID: P02686), which were shown to be engaged in the formation the non-amyloid protein aggregates, and which were considered as the major constituents defining preservation of the Heslington brain for at least 2,600 years [2].

In our view, these additions and clarifications included to the R1 version of the manuscript provided sufficient level of justification for calling neurofilament (NF) proteins (or NFs), glial fibrillary acidic protein (GFAP), and myelin basic protein (MBP) as “main proteins”. However, we changed warding in the first paragraph of the Results section to read:

At the first stage of our study, we looked at the intrinsic disorder predisposition of five “main proteins”, the neurofilament heavy, medium, and light chain proteins (NFH, NFM, and NFL, UniProt IDs: P12036, P07197, P07196, respectively), glial fibrillary acidic protein (GFAP; UniProt ID: P14136), and myelin basic protein (MBP; UniProt ID: P02686). These proteins are defined as “main” in this study based on the fact that they were shown to be engaged in the formation of the non-amyloid protein aggregates and were considered as the major constituents defining preservation of the Heslington brain for at least 2,600 years [2].

Taking all this information and all the changes which was already included into the R1 version of the manuscript, we cannot see what else can be added here to convince the reviewer that these proteins deserve special attention and can be called “main proteins”.

(2) statistical analysis, if I interpret correctly, does NOT show that this very old brain tissue has higher IDP content than the normal genome. So I don't see that the point is made that these IDPs are responsible for the preservation.

RESPONSE: We respectfully disagree with this comment as well. As a matter of fact, statistical analysis of the disorder levels in modern and ancient brain proteins revealed that the Heslington brain contains significant number of disordered proteins. This observation was already highlighted in the R1 version of the manuscript.

Once again, in the first review, this reviewer indicated:

(2) disordered proteins are not uncommon. Also, disordered proteins interacting with other IDPs is not uncommon. In order for the interactions to be important/noteworthy, the statistics of how many proteins by mass or by number are IDPs in the "old" tissue should be compared with the same analysis of "new" tissue. Perhaps this is discussed at some point, but it needs to be front and center, very clear, in the summary, abstract and conclusions. I don't see it in any of those places.

(3) In the absence of this sort of statistical justification, the presence of IDPs and their interactions is not a compelling argument for their importance for preservation.

These issues were addressed by conducting statistical analysis and including of the corresponding data and their discussion to the revised manuscript (pages 18-20). Furthermore, we also emphasized that the presence of very high proportion of moderately disordered proteins in the Heslington brain samples is very counterintuitive, as disordered proteins and regions are known to serve as primary targets for various proteases. As a result, many IDPs are characterized by shorter lifetimes than ordered proteins. The situation is different when IDRs or IDPs form complexes, many of which are highly intertwined and therefore very stable. As we indicated in the manuscript, it seems that the intrinsic disorder of NFH, NFM, NFL, GFAP, and MBP, their interactors and many other proteins might play a crucial role in preserving the Heslington brain by forming tightly folded brain protein aggregates, in which different parts are glued together via the disorder-to-order transitions. We further elaborated this subject and added some discussion of the related “stability of instability” phenomenon associated with intrinsic disorder in the R1 version.

In this new revision, we provide further discussion of these observations and indicate: “This conclusion is based both on the simple numerical analysis showing that more than 30% of the “Heslington brain proteins” are expected to be highly disordered and on the on the results of the statistical comparison of the modern brain proteome with the proteins found in the Heslington brain showing that the 2,600-year-old brain tissue is significantly enriched in intrinsically disordered proteins.”  
